# Structural determinants of Rab11 activation by the guanine nucleotide exchange factor SH3BP5

Meredith L. Jenkins[1], Jean Piero Margaria[2], Jordan T.B. Stariha[1], Reece M. Hoffmann[1], Jacob A. McPhail[1], David J. Hamelin[1], Martin J. Boulanger[1], Emilio Hirsch [2] & John E. Burke [1]

The GTPase Rab11 plays key roles in receptor recycling, oogenesis, autophagosome formation, and ciliogenesis. However, investigating Rab11 regulation has been hindered by limited molecular detail describing activation by cognate guanine nucleotide exchange factors (GEFs). Here, we present the structure of Rab11 bound to the GEF SH3BP5, along with detailed characterization of Rab-GEF specificity. The structure of SH3BP5 shows a coiled-coil architecture that mediates exchange through a unique Rab-GEF interaction. Furthermore, it reveals a rearrangement of the switch I region of Rab11 compared with solved Rab-GEF structures, with a constrained conformation when bound to SH3BP5. Mutation of switch I provides insights into the molecular determinants that allow for Rab11 selectivity over evolutionarily similar Rab GTPases present on Rab11-positive organelles. Moreover, we show that GEF-deficient mutants of SH3BP5 show greatly decreased Rab11 activation in cellular assays of active Rab11. Overall, our results give molecular insight into Rab11 regulation, and how Rab-GEF specificity is achieved.

[1] Department of Biochemistry and Microbiology, University of Victoria, Victoria, BC V8W 2Y2, Canada. [2] Department of Molecular Biotechnology and Health Sciences, Molecular Biotechnology Center, University of Turin, Via Nizza 52, 10126 Torino, Italy. Correspondence and requests for materials should be addressed to J.E.B.(email: jeburke@uvic.ca)

Critical to almost all aspects of membrane trafficking and cellular signaling is the ability to properly traffic membrane cargoes. Cells possess a highly regulated system for directing membrane cargo to the proper cellular location, with one of the key determinants being the regulation of Rab (**Ra**s related proteins in **b**rain) GTPases[1–4]. Rab proteins mediate specific exchange of proteins and lipid cargos between distinct intracellular organelles, through selective binding and recruitment of Rab-binding proteins. Thus, Rabs are crucial to proper cellular function and their dysregulation leads to a variety of disease states.

Rab GTPases act as molecular switches and cycle between a GDP "off" state and a GTP-bound "on" state. They are lipidated at the C terminus, with geranylgeranyl modifications of C-terminal cysteines mediating their membrane association. Human Rab proteins show a remarkable diversity, with over 66 identified members, of which at least 20 were present in the last common ancestor of all Eukaryotes[5]. The association of Rab-binding partners is specific to the nucleotide-bound state, with most Rab effectors having specificity for the GTP-bound active conformation. A majority of Rab GTPases have extremely low intrinsic rates of nucleotide exchange and GTP hydrolysis and require regulatory proteins to control their spatiotemporal activation and inactivation. The nucleotide-binding state of Rab GTPases is regulated through the coordinated interplay of activating guanine nucleotide exchange factors (GEFs) and inactivating GTPase-activating proteins (GAPs), with an additional level of control mediated by guanine nucleotide dissociation inhibitory proteins[6–9].

Among the best studied Rab GTPases are members of the Rab11 subfamily, which in humans comprised three isoforms (Rab11A, Rab11B, and Rab25 [also known as Rab11C]). The Rab11 proteins are master regulators of the surface expression of receptors[10]. They are primarily localized at the *trans*-Golgi network, post-Golgi vesicles, and the recycling endosome, and they facilitate cytokinesis[11], ciliogenesis[12], oogenesis[13], and neuritogenesis[14]. Intriguingly, Rab11-positive vesicles have also been identified as precursors for autophagosome assembly[15], which is one of the initiating steps of autophagy. Both Rab11A and Rab11B are mutated in developmental disorders, with putative inactivating Rab11A or Rab11B mutations leading to intellectual disability and brain malformation[16,17]. Overexpression of Rab25 has been implicated in poor prognosis in ovarian cancer, with stapled peptide inhibitors of Rab25-effector binding inhibiting migration and proliferation[18,19]. Many intracellular pathogens, including viruses[20], bacteria[21], and parasites[22] subvert membrane trafficking through targeting Rab11-positive vesicles.

Hindering the capability to fully understand Rab11 regulation is the lack of molecular detail describing how Rab11 proteins are activated by their cognate GEFs. Biochemical reconstitution of the large macromolecular TRAPPII complex with their cognate Rab11 homologs in both yeast (*Saccharomyces cerevisiae*) and fruit flies (*Drosophila melanogaster*) revealed clear GEF activity toward both Rab1 (yeast Ypt1) and Rab11 (yeast Ypt31/32)[23,24]. The *Drosophila* DENN protein Crag was also identified as a GEF toward both Rab10 and Rab11[25]. Neither of these GEFs is selective for Rab11, with both Rab10 and Rab1 having a different spatial organization compared with Rab11, implying the existence of other more specific Rab11 GEFs.

The first insight into potentially Rab11-specific GEF proteins was the discovery of the protein REI-1 and its homolog REI-2 in *Caenorhabditis elegans*, with loss of both leading to defects in cytokinesis[26]. These enzymes are found only in metazoans, and knockouts of the REI-1 ortholog in *Drosophila* (parcas, also known as poirot) are viable but have defects in oogenesis and muscle development[27–29]. Intriguingly, knockouts of either TRAPPII or parcas are viable in *Drosophila* but the double

knockout is embryonic lethal, suggesting the shared GEF activity for Rab11 is partially redundant[23]. The mammalian ortholog of REI-1, SH3-binding protein 5 (SH3BP5), also has GEF activity toward Rab11 and was shown to be selective for Rab11 over Rab5[26]. In addition, mammals contain a SH3BP5 homolog, SH3BP5L, which has not yet been tested for Rab11 GEF activity. The role of SH3BP5 in development and signaling is complicated by its additional capability to directly regulate Bruton tyrosine kinase (BTK) signaling through binding to the BTK SH3 domain[30], as well as to inhibit JNK signaling through engagement of the disordered C terminus of SH3BP5[31,32].

The fundamental molecular mechanism by which Rab11 proteins can be activated by their cognate GEFs has remained unclear. To decipher the mechanism of SH3BP5 GEF activity we have determined the structure of the GEF domain of SH3BP5 bound to nucleotide-free Rab11A. Detailed biochemical studies reveal that SH3BP5 is highly selective for Rab11 isoforms, with no activity toward any of the most evolutionarily similar Rab GTPases. A subset of clinical Rab11 mutations found in developmental disorders were found to disrupt SH3BP5-mediated nucleotide exchange, providing a possible mechanism of disease. Detailed mutational analysis of both Rab11 and SH3BP5 reveals the molecular basis for Rab selectivity, as well as allowing for the design of GEF-deficient SH3BP5 mutants. These SH3BP5 mutants were tested using a cellular Rab11 fluorescence resonance energy transfer (FRET) sensor and show significantly decreased Rab11 activation. Overall, our study thus reveals insight into Rab11 regulation and defines how Rab11-GEF selectivity is achieved.

## Results

**Biochemical characterization of SH3BP5 GEF activity**. SH3BP5 was previously demonstrated to act as a GEF for Rab11[26] and this activity was strongly dependent on the membrane presentation of Rab11. To examine the specificity of GEF activity for Rab11 family members, GEF assays were carried out on the Rab11 isoforms Rab11A and Rab25 loaded with the fluorescent GDP analog 3-(*N*-methyl-anthraniloyl)-2-deoxy-GDP (Mant-GDP) and nucleotide exchange was determined as a function of SH3BP5 concentration. Rab11 proteins were generated with a C-terminal His-tag, which allows for localization on NiNTA-containing membranes (Fig. 1a). Domain schematics of all purified protein constructs generated in this study are shown in Supplementary Figure 1.

The catalytic efficiency of SH3BP5 GEF activity ($k_{cat}/K_m$) was highest for Rab11A, $\sim 3.5 \times 10^4\,M^{-1}\,s^{-1}$, with slightly lower values for Rab25 (11c) at $\sim 1.8 \times 10^4\,M^{-1}\,s^{-1}$ (Fig. 1b). SH3BP5L GEF activity was even higher, with values of $\sim 8.1 \times 10^4\,M^{-1}\,s^{-1}$ against Rab11A. Measurements of SH3BP5 Rab GEF activity were characterized both in solution and in a membrane-reconstituted system where Rab isoforms were attached to NiNTA-containing membrane using a C-terminal His-tag, similar to previous Rab11 GEF studies on the *Drosophila* and yeast variants of the TRAPPII complex[23,24]. SH3BP5 GEF activity showed only a weak dependence on Rab11A being present on a lipid membrane, with a $\sim 2$-fold higher rate on a membrane compared with free in solution, consistent with biochemical studies performed on the *Drosophila* homolog parcas[23].

Previously, Rab11 activation has been associated with the generation of different phosphoinositides, including phosphatidylinositol-3-phosphate (PI3P) and L-α-phosphatidylinositol-4-phosphate (PI4P)[33–35]. To test this biochemically, GEF assays were carried out on vesicles of both different size (100 nm and 400 nm) and different membrane composition including variations of surface charge and phosphoinositide (PS, PI, PI3P,

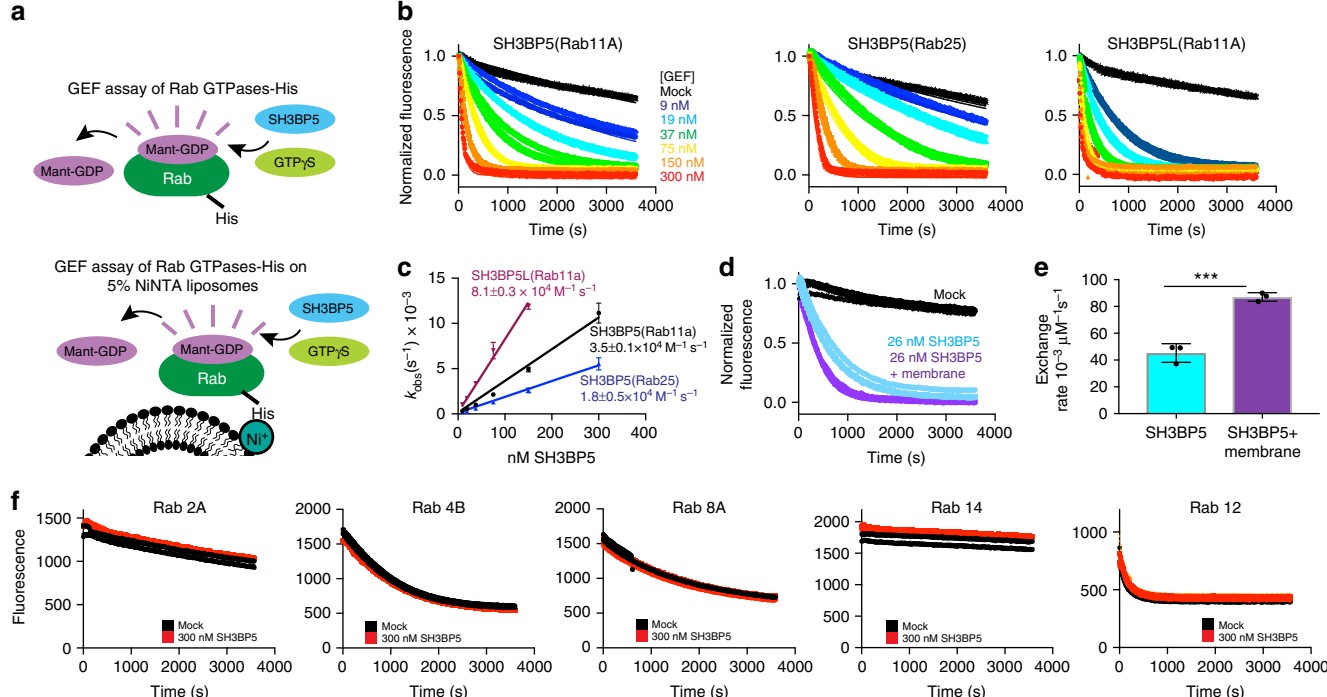

**Fig. 1** In vitro GEF assays reveal that SH3BP5 is a potent and selective GEF for Rab11. **a** Schematic of GEF activation assays using fluorescent analog Mant-GDP in the presence and absence of NiNTA-containing lipid vesicles. **b** In vitro GEF assay of SH3BP5 (1–455) on Rab11A and Rab25, and SH3BP5L (1–393) on Rab11A. Nucleotide exchange was monitored by measuring the fluorescent signal during the SH3BP5(1–455) (9 nM-300 nM) catalyzed release of Mant-GDP from 4 μM of Rab11a-His$_6$ in the presence of 100 μM GTPγS. Each concentration was conducted in triplicate. **c** Calculation of $k_{cat}/K_m$ values for nucleotide exchange rates of Rab11A (SH3BP5 and SH3BP5L) and Rab25 (SH3BP5) according to the protocol of ref. [61]. **d** In vitro GEF assay of SH3BP5 on Rab11A in the presence and absence of 100 nm extruded liposomes at 0.2 mg/ml (10% DGS-NTA(Ni), 10% PS, 15% PE, 20% PI, 45% PC). Nucleotide exchange was monitored by measuring the fluorescent signal during the SH3BP5 (1–455) (26 nM)-catalyzed release of Mant-GDP from 4 μM of Rab11A-His$_6$ in the presence of 100 μM GTPγS. **e** Bar graph representing the difference in GEF activity in the presence and absence of Ni-NTA membranes. **f** In vitro GEF assays of SH3BP5 against a panel of evolutionarily related Rab GTPases loaded with Mant-GDP at a final concentration of 300 nM SH3BP5 and 4 μM Rab GTPase. For all panels, error bars represent SD ($n = 3$), with $p$-values generated using a two-tailed Student's $t$-test (***$p < 0.005$)

and PI4P) composition (Supplementary Figure 2a). There was no difference in membrane-stimulated GEF activity across all membranes tested (Supplementary Figure 2b). Rab GEF-mediated nucleotide exchange has also been previously shown to have a strong dependence on dominant active switch II mutants[36] (Q70L in Rab11). However, GEF assays on Q70L Rab11A showed a slightly elevated GEF-mediated nucleotide exchange rate for Q70L compared with wild-type (WT) Rab11A (Supplementary Figure 2d). The dominant negative mutant of Rab11A (S25N) caused rapid release of Mant-GDP even in the absence of SH3BP5 (Supplementary Figure 2d).

SH3BP5 has been found to be selective for Rab11 over Rab5[26]. However, these GTPases are highly evolutionarily distinct. To define the selectivity of SH3BP5 for Rab11, we characterized the GEF activity of SH3BP5 toward the most evolutionarily similar Rab GTPases (Rab2A, Rab4B, Rab14, and Rab12)[5] as well as Rab8, which is activated downstream of a Rab11-dependent Rab cascade[12]. SH3BP5 showed no detectable Rab GEF activity for any of these Rab GTPases at up to 300 nM concentration of GEF, revealing that SH3BP5 is both a highly potent and highly selective Rab11 GEF (Fig. 1f).

**Structure and dynamics of the Rab11A–SH3BP5 complex.** To define the dynamics of the full-length SH3BP5 protein, we used hydrogen deuterium exchange mass spectrometry (HDX-MS). HDX-MS measures the exchange rate of amide hydrogens with solvent, and as the main determinant of exchange is the stability of secondary structure, it is an excellent probe of secondary structure dynamics[37]. HDX experiments with extremely short

exposures of D$_2$O can be used to identify disordered regions within proteins[38]. We carried out HDX experiments with a short pulse of deuterium exposure (3 s at 1 °C) for the full-length SH3BP5, with both the N terminus and C terminus of SH3BP5 having >50% deuterium incorporation, indicating limited secondary structure (Supplementary Figure 3a). HDX-MS results were used to generate a crystal construct of SH3BP5, with all of the C-terminal disordered region removed (SH3BP5 1–265). GEF assays carried out using either full-length SH3BP5, a N-terminally truncated (31–455) version, and the 1–265 crystal construct revealed no difference in SH3BP5-mediated Rab11A nucleotide exchange, suggesting that the disordered N and C termini have no effect on Rab11 binding or SH3BP5 GEF activity (Supplementary Figure 3c, d). The complex of the crystal construct of SH3BP5 (1–265) and Rab11A Q70L was able to be purified to homogeneity and eluted as a 1:1 complex on gel filtration (Supplementary Figure 3e, f).

To understand the molecular basis of how SH3BP5 mediates Rab11A nucleotide exchange, we determined the structure of the N-terminal GEF domain of SH3BP5 bound to full-length Q70L Rab11A to a final resolution of 3.1 Å. The Q70L construct of Rab11 was used for structural studies as the marginally increased GEF rate for Q70L vs. WT suggested Q70L might form a slightly more stable complex with SH3BP5. Initial phases were generated by SeMet single wavelength anomalous diffraction of 3.3 Å and extended to 3.1 Å using native data (Fig. 2, details on unit cell data collection and refinement details are in Supplementary Figure 4 and Table 1). The structure is composed of four complexes of SH3BP5-Rab11A per asymmetric unit, with each of

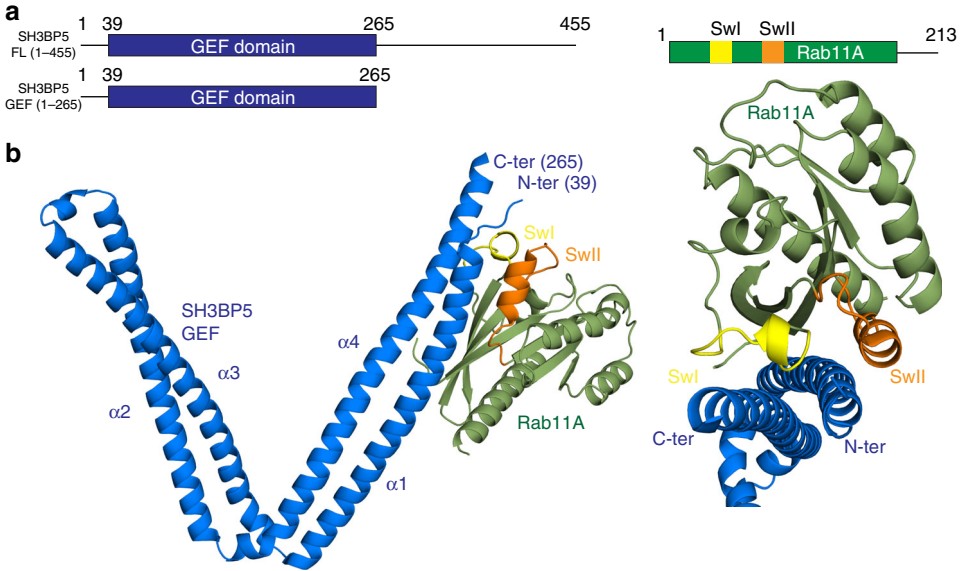

**Fig. 2** Structure of SH3BP5 in complex with Nucleotide-free Rab11. **a** Schematic of full-length SH3BP5(1–455), the crystal construct of SH3BP5 (1–265) and Rab11A with switch I (SwI) and switch II (SwII) regions annotated. **b** The structure of the GEF domain of SH3BP5 with nucleotide-free Rab11 solved to 3.1 Å resolution. The GEF domain of SH3BP5 is composed of a v-shaped coiled coil with four long α-helices annotated on the figure. The switch regions of Rab11 are colored yellow (Switch I (SwI)) and orange (switch II (swII)), with the SwI situated between helix α1 and α4 of SH3BP5

| Table 1 Data collection and refinement statistics | | |
|---|---|---|
| | **SH3BP5-Rab11 native** | **SH3BP5-Rab11 SeMet (peak)** |
| **Data collection** | | |
| Space group | I222 | I222 |
| Cell dimensions | | |
| $a, b, c$ (Å) | 114.85, 197.04, 304.71 | 114.13, 197.18, 304.20 |
| $\alpha, \beta, \gamma$ (°) | 90, 90, 90 | 90, 90, 90 |
| Resolution (Å) | 49.26–3.10 (3.21–3.10)[a] | 49.39–3.25 (3.36–3.25) |
| $R_{merge}$ | 0.1363 (2.189) | 0.156 (2.214) |
| $I / \sigma I$ | 8.26 (0.75) | 13.68 (1.35) |
| Completeness (%) | 99.74 (99.79) | 99.88 (99.72) |
| Redundancy | 4.9 (5.1) | 13.2 (13.0) |
| **Refinement** | | |
| Resolution (Å) | 49.26–3.1 (3.21– 3.1) | |
| No. reflections | 62,902 (6236) | |
| $R_{work} / R_{free}$ | 23.5/27.8 | |
| No. atoms | | |
| Protein | 12,888 | |
| Ligand/ion | 0 | |
| Water | 0 | |
| $B$-factors | | |
| Protein | 132.44 | |
| Ligand/ion | | |
| Water | | |
| R.m.s. deviations | | |
| Bond lengths (Å) | 0.003 | |
| Bond angles (°) | 0.55 | |

[a]Values in parentheses are for highest-resolution shell.
Number of crystals used for structure = 1

the copies in the asymmetric unit sharing an overall similar architecture. The following structural analysis refers to the complex of SH3BP5 chain E and Rab11A chain F, as this complex showed the best-defined electron density. The GEF domain of SH3BP5 is composed of a coiled coil that is kinked to form an overall v-shape. The v is composed of four long α-helices (α1–α4), with a small helix connecting helices α2 α3. This connecting region between helix α2 α3 shows the highest degree of conformational flexibility between copies in the asymmetric unit (Supplementary Figure 4). The putative SH3-binding site for BTK is located at the turn from helix α3 to α4[30]. However, this region shares no similarity to any previously identified SH3-binding site. A surface potential map of SH3BP5 reveals limited highly charged regions, with one basic stretch located at the base of the v shape (Supplementary Figure 4). There also are no putative amphipathic helices in SH3BP5 that would potentially mediate membrane binding. The N-terminal GEF domain of SH3BP5 was previously predicted to share homology to F-bar domains[26]. However, bioinformatic analysis using the DALI server revealed that the closest structural homologs are the cell invasion protein SipB (PDB: 3tul [https://www.rcsb.org/structure/3tul]), the stalk region of dynein (PDB: 5ayh [https://www.rcsb.org/structure/5ayh]), and the bacterial chaperone prefoldin (PDB: 1fxk [https://www.rcsb.org/structure/1fxk]). The highest similarity to the α1/α4 Rab11 binding coiled coil is the archaeal chaperone prefoldin[39]. The characteristic v-shape of SH3BP5, however, is unique among solved coiled-coil proteins.

Nucleotide-free Rab11A was bound to the GEF domain of SH3BP5 at one end of the v at a surface composed of the N terminus of helix α1 and the C terminus of helix α4 of the coiled coil. The SH3BP5-Rab11 interface is composed of a large extended, primarily hydrophobic interface (~1250 Å²) (Fig. 3a). The Rab11-binding surface of SH3BP5 is composed of residues 39–67 and residues 240–262. The contact residues at this interface are highly conserved both in SH3BP5 and SH3BP5L, as well as the parcas and REI-1/2 homologs in both *Drosophila* and *C. elegans*, respectively (Fig. 3b). The SH3BP5-binding surface of Rab11 is composed of residues spanning the N terminus (residues 6–13), residues in and near switch I (residues 35–48), the inter-switch region 58–65, residues in and near switch II (72–85), and the C terminus (180–181). Rab11 packs along the center of the coiled coil of SH3BP5, with the main binding interface composed of a contiguous hydrophobic surface of Rab11 residues including Y8, L10, and Y11 at the N terminus, F36 and I44 in switch I, Y73

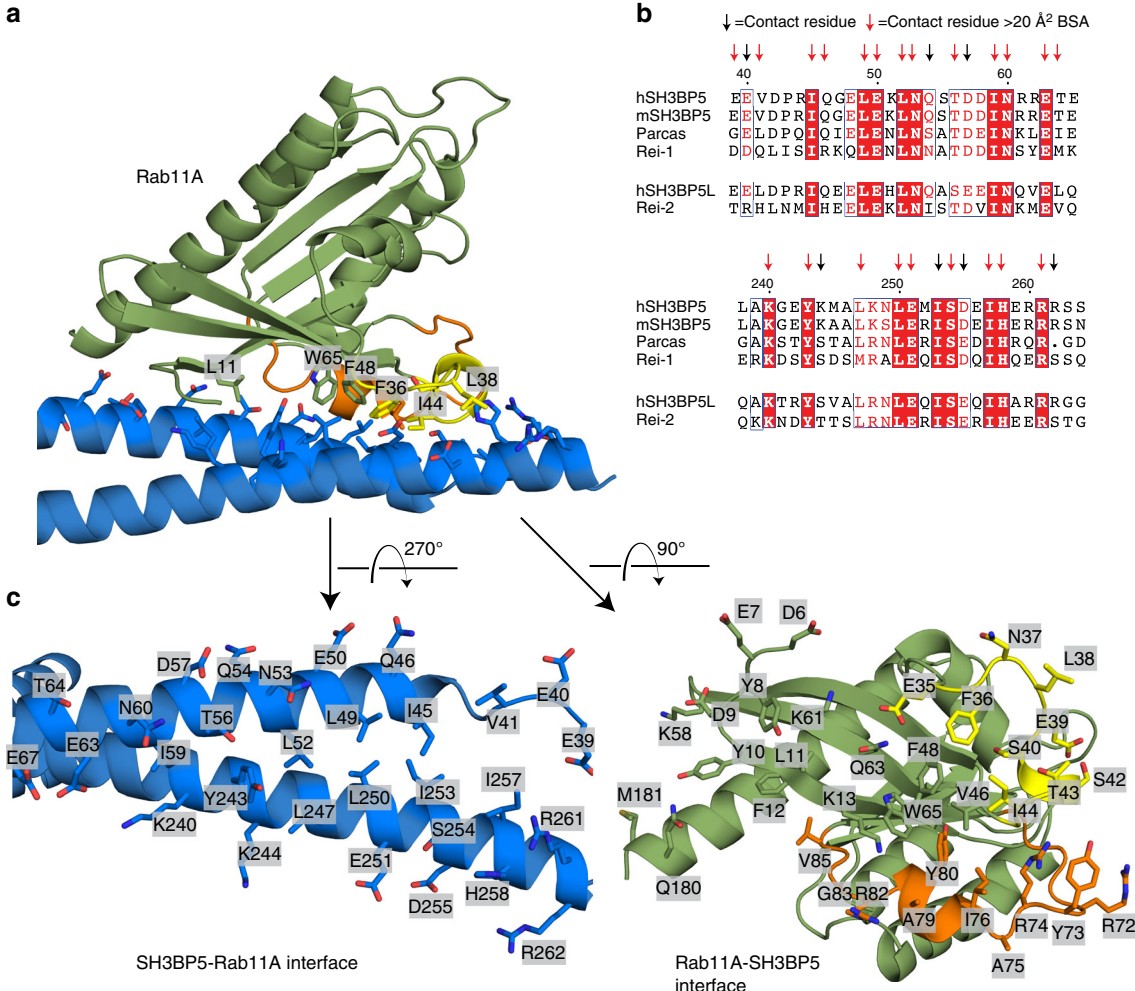

**Fig. 3** Rab11A SH3BP5 interface. **a** Zoom in on the binding interface of Rab11 and SH3BP5. Key Rab11 residues forming important interactions at the interface are labeled. **b** Alignment of the Rab11-binding site on both helix α1 and α4 of SH3BP5 and SH3BP5L along with paralogs from *M. musculus* (mSH3BP5), *D. melanogaster* (parcas), and *C. elegans* (REI-1/REI-2), with contact residues indicated by arrows. Red arrows indicate greater than 20 Å² of buried surface area (BSA). **c** All interacting residues from both SH3BP5 and Rab11 are shown in stick representation and labeled, with the structures rotated and the interacting partner removed to allow for visualization. SwI and SwII are colored according to Fig. 2

and I76 in switch II, and the hydrophobic triad (F48, W65, and Y80), which has a conserved role in interacting with Rab effectors[40] (Fig. 3c).

HDX-MS experiments were carried out to both verify the crystallographic complex as well as define how SH3BP5 works on a membrane surface. HDX-MS experiments revealed differences in exchange for Rab11A Q70L when bound to a N-terminally truncated SH3BP5 (31–455), with decreased HDX at the SH3BP5 interface (Supplementary Figure 5a–c), verifying the crystallographic interface in solution. Increased exchange was observed surrounding the Rab11 nucleotide-binding pocket and this likely corresponds to loss of nucleotide upon SH3BP5 binding. HDX-MS experiments mapping out the Rab11 interface for the N-terminally truncated variant of SH3BP5 (31–455) were carried out with both soluble and membrane localized WT Rab11A (1–211, C-term his) to analyze conformational changes that may occur upon membrane binding. Decreases in exchange were observed for SH3BP5 at the crystallographic interface with Rab11A, with the only consequence of membrane being a larger decrease in exchange in Rab11A interfacial residues in SH3BP5 (Supplementary Figure 5d–f) and no additional regions showing changes in H/D exchange. This potentially signifies a tighter

interaction between SH3BP5 and Rab11A occurring on a membrane surface, which might possibly explain the limited selectivity of SH3BP5 for different membrane compositions (Supplementary Figure 2).

**Comparison with previously solved Rab-GEF complexes.** Comparison of the structure of nucleotide-free Rab11A bound to SH3BP5 compared with Rab11A bound to either GDP or GTP reveals an extensive rearrangement of switch I[41], with only limited conformational changes in switch II (Fig. 4a and Supplementary Figure 6, Supplementary Movie 1). One of the minor conformational changes that occurs in switch II upon SH3BP5 binding is a reorientation of Y73, where the hydroxyl group is in position to cap the switch I helix (S40-G45). There were also conformational changes that occurred around the nucleotide-binding pocket, including decreased secondary structure, although not to the same extent as seen in the Rab8:MSS4 GEF complex[42]. HDX-MS studies of differences between nucleotide-bound Rab11 and SH3BP5-bound nucleotide-free Rab11 showed large increases in exchange in all regions of Rab11 around the nucleotide-binding pocket when bound to SH3BP5

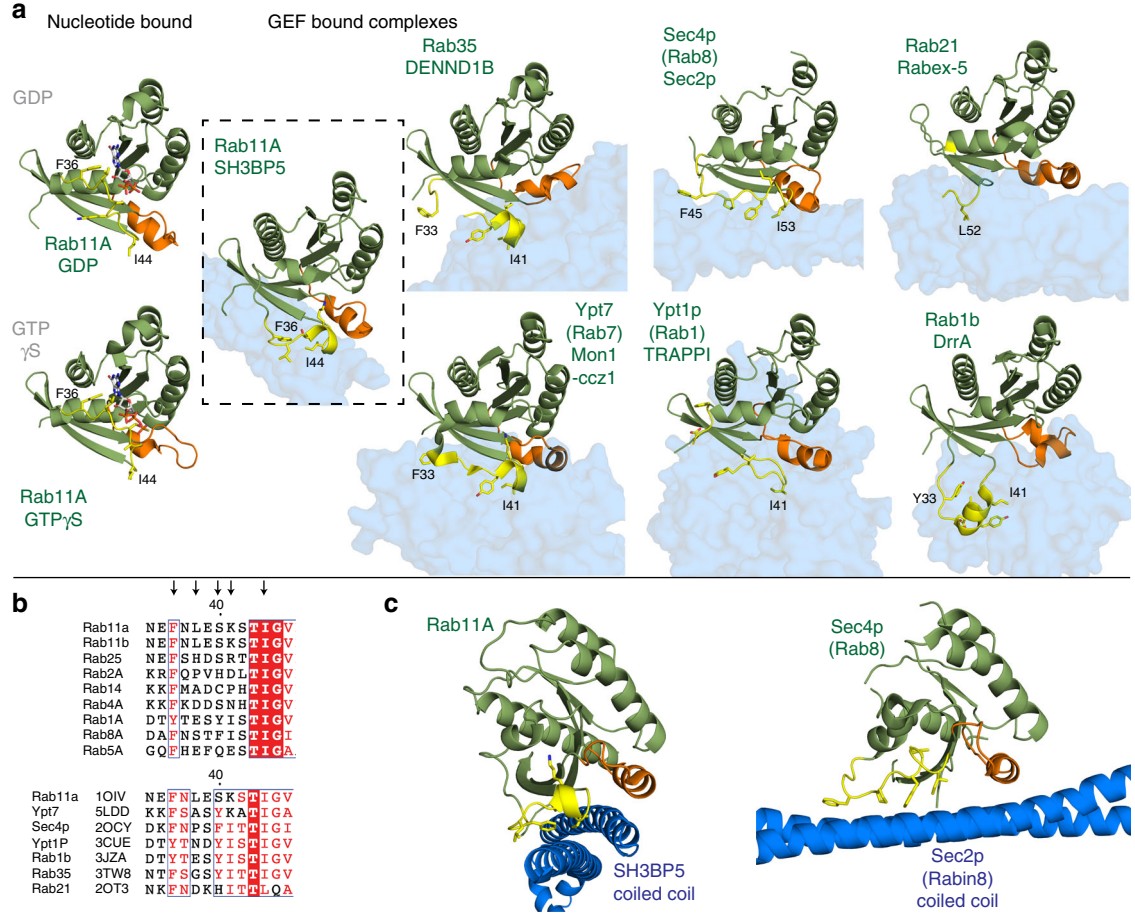

**Fig. 4** Unique switch orientations of Rab11-SH3BP5 compared to previously solved Rab-GEF structures. **a** Structures of nucleotide-bound Rab11 are compared to the GEF bound structures of Rab11-SH3BP5 (dotted box), Rab35-DENND1B[43], Ypt7-Mon1-Ccz1[44], Sec4p-Sec2p[47], Ypt1p-TRAPPI[52], Rab21-Rabex-5[70], and Rab1b-DrrA[56]. The Rab GTPases are aligned to each other, with the GEF domain shown as a transparent surface. Switch I and II are colored according to Fig. 2, and the residues corresponding to F36 residue and I44 in Rab11 are labeled on all structures (the residue corresponding to F36 is disordered in the Ypt1p-TRAPPI complex, and in the Rab21-Rabex-5 structures). **b** Alignment of the Switch I of Rab11 to a selection of closely related Rab isoforms and all previously solved Rab-GEF structures (PDB codes indicated). Switch I residues shown as sticks are highlighted by arrows. **c** Alignment of the coiled coil GEF domains of SH3BP5 and Sec2P (Rabin8)[47]. This reveals a completely orthogonal binding interface, with the two GEF coiled coils binding to their cognate Rab in a perpendicular orientation, revealing independent evolution of coiled coils as Rab-GEFs

(Supplementary Figure 5a–c), further highlighting decreased stability of the nucleotide-binding pocket.

The substantial conformational change of switch I upon SH3BP5 binding is present in all previously solved Rab-GEF structures and is predicted to mediate nucleotide exchange through disrupting switch I nucleotide interactions. However, comparison of the conformational change in switch I reveals striking differences between previously solved Rab-GEF structures. The main difference is in the conformation of F36 and I44 in Rab11. These residues pack directly against each other, which requires a constrained conformation of switch I. This constrained orientation of the highly conserved F36 and I44 residues (Fig. 4a, b) is unique compared with previously solved Rab-GEF structures. There are a number of similarities with previously solved GEF structures, with a helix forming from S40 to G45 in switch I of Rab11, similar to DENND1B-Rab35[43] and Mon1-Ccz1-Ypt7[44] structures. Lysine 41 in Rab11 when bound to SH3BP5 is oriented toward the empty $Mg^{2+}$-binding site; however, there was limited electron density around this residue, indicating that it is quite flexible (Supplementary Figure 4). This same lysine residue in Ypt7 has a key role in activation by Mon1-Ccz1[44], with the mechanism proposed to be mediated through disruption of the $Mg^{2+}$-binding site.

Individual point mutations in Rab11A (K13N, K24R, R82C, and S154L) and Rab11B (V22M and A68T) are found in developmental disorders that lead to intellectual disability[16,17]. Mapping these mutations onto the structures of nucleotide-bound Rab11, Rab11-GTPγS bound to the Rab11 effector FIP3, and SH3BP5-Rab11 revealed possible roles of these mutations in Rab11 regulation (Fig. 5, Supplementary Figure 7). The V22M and S154L would be expected to sterically disrupt nucleotide-binding and HDX-MS experiments of these mutants compared with WT Rab11 revealed large increases in exchange throughout the entire Rab protein (Supplementary Figure 7), indicating that indeed these mutants are not able to bind nucleotide. No GEF assays were able to be carried out on these mutants, as they could not be loaded with Mant-GDP. K24R is part of the P-loop that coordinates the phosphate groups in bound nucleotide and GEF assays revealed that this mutant had rapid nucleotide exchange even in the absence of SH3BP5 (Fig. 5), indicating decreased affinity for nucleotide. HDX-MS experiments on this mutant revealed increased exchange throughout the majority of Rab11, although not as large as changes seen for V22M and S154L (Supplementary Figure 7). The A68T mutant is located at the beginning of switch II, with the Ala oriented toward the region of switch II that interacts with SH3BP5, and it only showed a modest

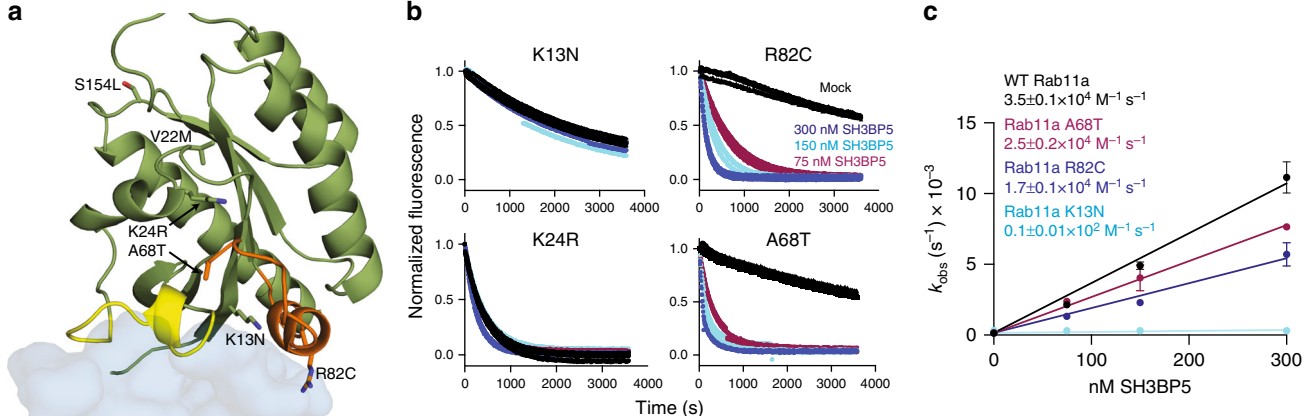

**Fig. 5** Clinically relevant Rab11 mutations disrupt nucleotide-binding or alter SH3BP5 GEF activity. **a** Point mutations found in developmental disorders mapped on the structure of Rab11 bound to SH3BP5. **b** In vitro GEF assay of SH3BP5 on clinical Rab11 mutations. Nucleotide exchange was monitored by measuring the fluorescent signal during the SH3BP5 (1–455) (300 nM, 150 nM, or 75 nM) catalyzed release of Mant-GDP from 4 μM of the indicated Rab in the presence of 100 μM GTPγS. Fluorescent measurements were completed every 11 s for a total of 60 min (Excitation $\lambda = 366$ nm; Emission $\lambda = 443$ nm). **c** Nucleotide exchange rates of Rab11A mutants plotted as a function of SH3BP5 concentration. The $k_{cat}/K_m$ values for all Rab11A mutations were calculated from the slope. Error bars represent SD ($n = 3$).

effect in decreasing SH3BP5-mediated GEF activity. K13N and R82C are located directly at the Rab11-SH3BP5 interface, with R82 forming a salt bridge with E50 in SH3BP5 and K13 forming a polar bond with Q53 in SH3BP5. The K13 residue also forms pi-stacking interaction with the hydrophobic triad residue W65, which directly binds to SH3BP5. Mutation of K13N resulted in a > 100-fold decrease in SH3BP5-mediated GEF activity, with R82 only partially decreasing SH3BP5 GEF activity (Fig. 5). However, these mutations are unlikely to modify only GEF activity, as the R82 residue also forms a salt bridge with E747 in FIP3[45].

The discovery that the SH3BP5 coiled coil acts as a GEF suggested that it may act similarly to the previously structurally characterized coiled coil Sec2p GEF Sec4p (Rab8 and Rabin8 in humans)[46,47]. Surprisingly, the SH3BP5 coiled coil is perpendicular to the Sec2p coiled coil (Fig. 4c) and the two Rab GEFs interact with completely divergent binding surfaces on their cognate Rabs. Intriguingly, this suggests independent evolution of coiled coils to act as GEFs toward Rab GTPases.

**Defining the molecular basis of SH3BP5 Rab11 selectivity.** To further understand how SH3BP5 is able to achieve selectivity for Rab11 over evolutionarily similar Rab GTPases, we carried out site-directed mutagenesis of switch I residues in Rab11. We tested the role of F36 and I44 residues, with mutation of these residues to Alanine leading to almost completely abrogated SH3BP5-mediated GEF activity (Fig. 6a–c). Mutations in switch I that mimic residues found in Rab2A (L38P), Rab8 (S40F), and Rab14 (K41P) were also found to decrease GEF activity > 100-fold, with limited effect on intrinsic nucleotide exchange (Fig. 6a, b). These residues would all be predicted to disrupt the constrained conformation of F36 and I44 relative to each other, leading to disruption of the contiguous hydrophobic surface required for interaction with SH3BP5. Mutation of these residues to Ala, also greatly decreased GEF activity. In Ypt7, the yeast homolog of Rab7, K41 has a key role in mediating GEF activity through insertion into the $Mg^{2+}$-binding site. To determine whether this was conserved in Rab11, we tested GEF activity of a K41A Rab11 mutant. This led to a relatively minor ~3-fold decrease in GEF activity, compared with the previously reported 10-fold decrease in GEF activity for Mon1-Ccz1 on the K38A mutant of Ypt7[44]. This suggests that the role of this lysine residue in inserting into the $Mg^{2+}$-binding site is less important in SH3BP5-mediated GEF activity on Rab11 compared with Mon1-Ccz1-mediated GEF activity toward Ypt7.

**Generation of SH3BP5 GEF-deficient mutations.** Critical to understanding the role of SH3BP5 is to define differences between its BTK/JNK regulatory roles and Rab11 regulatory roles, which requires the generation of point mutations that disrupt only specific SH3BP5 functions. From examining the structure of SH3BP5 bound to Rab11 and sequence conservation throughout evolution, we separately mutated Rab11 contact interfaces in helix α1 (LNQ52AAA) and α4 (LE250AK) of SH3BP5. To ensure the protein was properly folded, we used HDX-MS to test conformational dynamics. Both mutants showed increased exchange only at the Rab11 interface, with no global changes in deuterium incorporation (Supplementary Figure 7g), suggesting these mutants remain properly folded. Both mutants led to a > 100-fold decrease in GEF activity compared with WT SH3BP5 (Fig. 6d, e).

To study activation of Rab11 in a cellular context, we used a recently developed Rab11 activation sensor (AS-Rab11)[33]. AS-Rab11 is composed of the Rab-binding domain (RBD) of FIP3, monomeric yellow fluorescent protein (mcpVenus), a proteinase K-sensitive linker, a monomeric cyan fluorescent protein, and human Rab11A (Fig. 7a). The RBD of FIP3 is specific to GTP-bound Rab11, with the consequence being that in the absence of active Rab11 there is low FRET and upon Rab11 activation the FRET signal increases. This was verified using AS-Rab11 mutants, both active Rab11 (AS-Rab11 Q70L) and inactive Rab11 (AS-Rab11 S25N), with increased FRET levels for Q70L, and decreased FRET for S25N (Supplementary Figure 8).

Immunofluorescence of FLAG-tagged SH3BP5 revealed an intracellular membrane distribution (Fig. 7b–d) similar to the AS-Rab11 sensor (Fig. 7c). Although SH3BP5[wt] showed an important colocalization with both total Rab11 and active Rab11 (Fig. 7d, e, Supplementary Figure 8B), the two GEF mutants SH3BP5[AAA] and SH3BP5[AK] displayed lower colocalization with active Rab11 (Fig. 7d, e). Accordingly, transfection of SH3BP5 led to increased Rab11 activation, which was not observed with transfection of the GEF-deficient mutants, confirming that these mutations disrupt Rab11 activation by SH3BP5 in living cells and cell lysates (Fig. 7f, Supplementary Figure 8c, d).

## Discussion

Rab11 GTPases are essential mediators of numerous membrane-trafficking processes and are especially important in controlling receptor recycling. Most Rab11-dependent processes have been

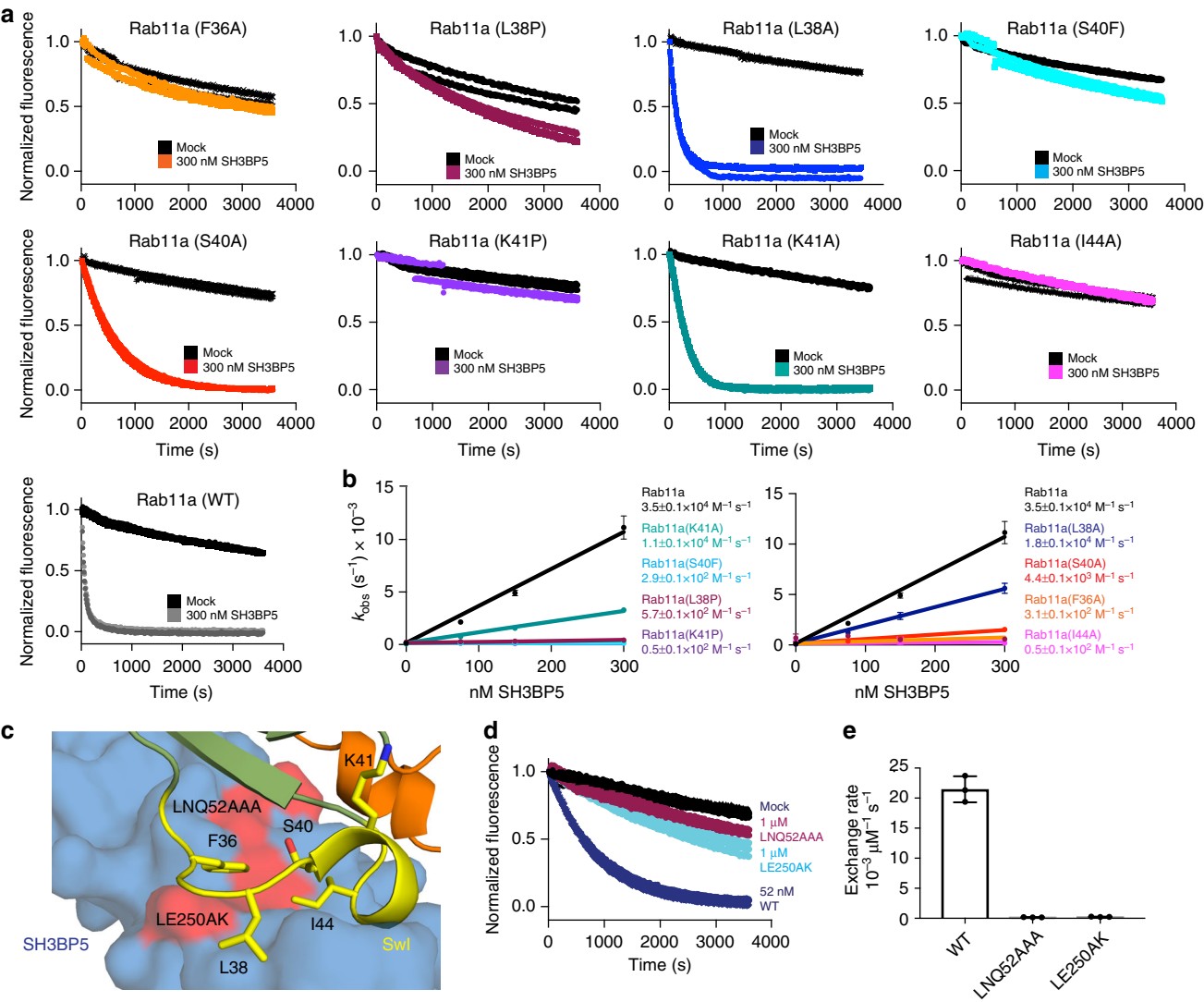

**Fig. 6** Molecular basis of SH3BP5 specificity and generation of GEF-deficient mutants. **a** In vitro GEF assay of SH3BP5 on Switch I Rab11 mutations. Nucleotide exchange was monitored by measuring the fluorescent signal during the SH3BP5 (1–455) (300 nM) catalyzed release of Mant-GDP from 4 µM of the indicated Rab in the presence of 100 µM GTPγS. Fluorescent measurements were completed every 11 s for a total of 60 min (Excitation $\lambda = 366$ nm; Emission $\lambda = 443$ nm). Two GEF curves are shown for each condition. **b** Nucleotide exchange rates of Rab11A mutants plotted as a function of SH3BP5 concentration. The $k_{cat}/K_m$ values for all switch I Rab11A mutations were calculated from the slope. All points have error bars, some are smaller than the size of the point. **c** Zoomed in view of the binding interface of SH3BP5 and Rab11. Key Rab11 residues involved in the interface with SH3BP5 are shown as sticks and labeled on the structure. Regions in SH3BP5 critical for GEF activity (LNQ52, LE250) are highlighted in red. **d** GEF assay of WT SH3BP5 (52 nM) and GEF-deficient mutants SH3BP5 (LNQ52AAA) and SH3BP5(LE250AK) (1000 nM), in the presence of the same concentrations of Rab11A and GTPγS as described in **a**. Two GEF curves are shown for each condition. **e** Quantification of Rab11 GEF activity for WT SH3BP5 and GEF-deficient mutations. For all panels error bars represent SD ($n = 3$)

studied through the use of dominant active (Q70L) and dominant inactive (S25N) mutants. These mutations can lead to misleading results, as many Rab GTPases are activated by GEFs through direct interactions with the catalytic Gln residue[36], leading to impaired activation for GTPase-deficient mutants. Complicating the ability to fully probe Rab11 and its role in both fundamental biological processes and disease has been the lack of molecular detail for the coordinated action of Rab11 GEFs and GAPs. Multiple Rab11 GAPs have been identified, including the proteins TBC1D11[48], TBC1D15[49], and EVI5[50,51]. Similarly, multiple Rab11 GEFs have been characterized, with the yeast and *Drosophila* TRAPPII complex activating both Rab1 and Rab11[23,24], and the *Drosophila* DENN family protein Crag being identified as a weak activator of Rab11, with a much higher activity towards Rab10[25]. SH3BP5, which is conserved in *C. elegans*, is a potent

Rab11 GEF[26] and knockout of the *Drosophila* homolog parcas leads to defects in oogenesis[27]. Intriguingly, it has been shown that Rab11 has a key role in oogenesis, through mediating asymmetric cell division[13], revealing a potential role of SH3BP5 in Rab11 regulation.

The structure of SH3BP5 bound to Rab11 reveals that the extended coiled coil of the GEF domain forms an extended predominantly hydrophobic interface with Rab11. The catalytic efficiency of SH3BP5L (~$8.1 \times 10^4$ M$^{-1}$ s$^{-1}$) and SH3BP5 (~$3.5 \times 10^4$ M$^{-1}$ s$^{-1}$) is comparable to previously determined Rab GEFs, including the Mon1-Ccz1 complex (~$2.1 \times 10^4$ M$^{-1}$ s$^{-1}$)[44], DENND1 (~$2.9 \times 10^4$ M$^{-1}$ s$^{-1}$)[43], and the TRAPP complex (~$1.6 \times 10^4$ M$^{-1}$ s$^{-1}$)[52]. There was a slight activation of SH3BP5 when Rab11 was presented on a membrane surface, with no strong dependency on surface charge, or phosphoinositide

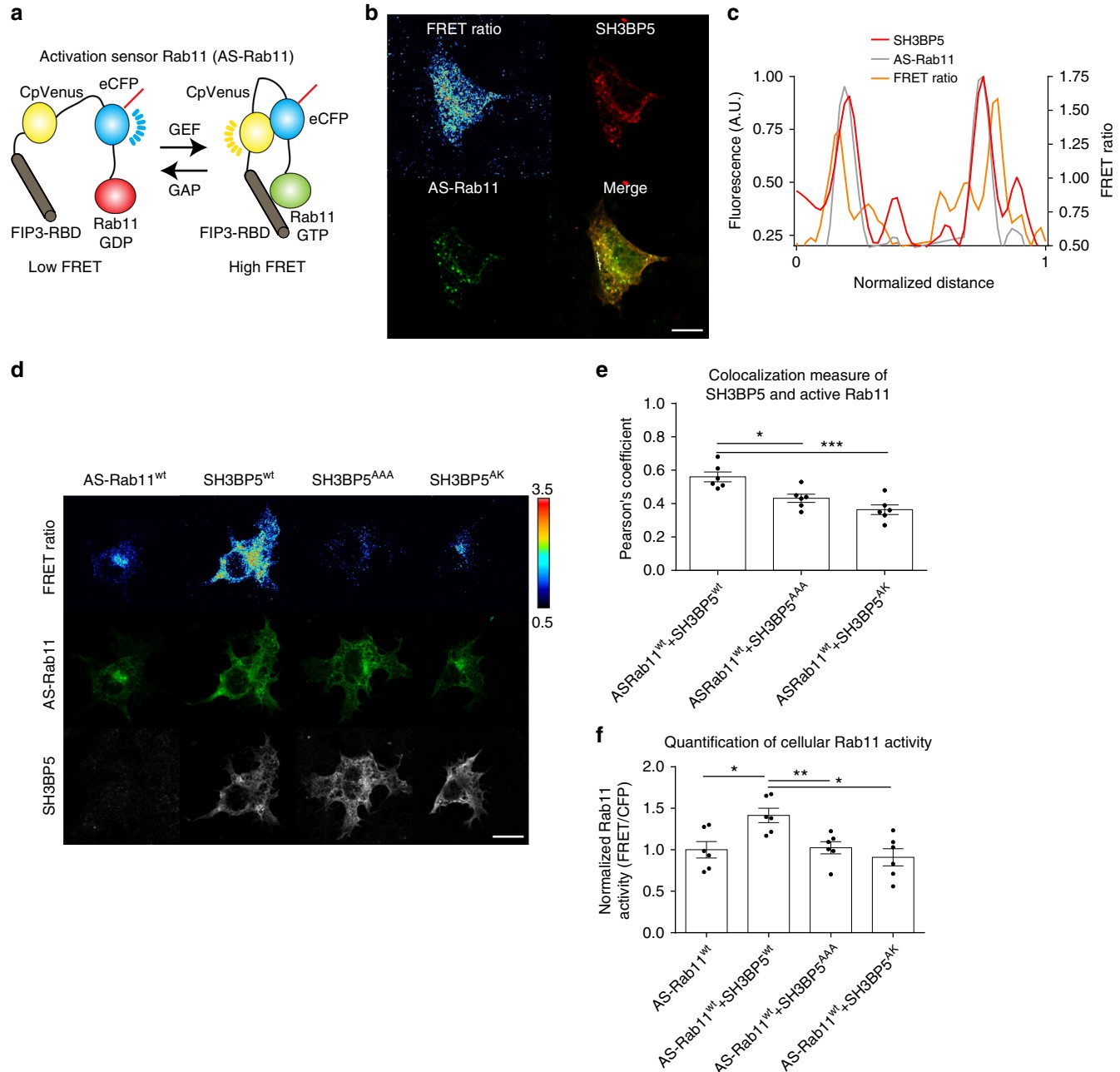

**Fig. 7** Cellular assays of Rab11 activation. **a** Schematic of activation sensor Rab11 (AS-Rab11)[33]. Upon activation by GEF proteins, GTP-loaded Rab11 will bind to FIP3 inducing FRET between the two fluorescent proteins. **b** Immunofluorescence localization of FLAG-tagged SH3BP5 and the AS-Rab11 sensor, along with the FRET ratio, measuring active Rab11. Scale bar, 10 μm. **c** Representative line intensity profile of AS-Rab11, SH3BP5, and the active pool of Rab11 (FRET ratio). **d** FRET ratio and localization of AS-Rab11 alone or with SH3BP5^wt, as well as the two GEF-deficient mutations (LNQ52AAA and LE250AK). Right side bar represents upper and lower limits of the FRET/CFP ratio. Scale bar, 10 μm. **e** Quantification of colocalization of SH3BP5 and active Rab11 (FRET ratio). **f** Quantification of Rab11 activity in COS-7 cells transfected with AS-Rab11 alone or in concomitance with SH3BP5^wt, SH3BP5^AK, or SH3BP5^AAA. For all panels, error bars represent SEM ($n = 6$). Significance determined by one-way ANOVA (*$p < 0.05$, **$p < 0.01$, ***$p < 0.005$)

content. Rab11 activation has been found to be associated with different phosphoinositide species, including PI4P during cytokinesis[35] and PI3P in ciliogenesis[34]. One of the main ways that Rab proteins are localized is through targeting of GEFs to specific intracellular membranes, and further experiments will be needed to define if SH3BP5 is involved in phosphoinositide-dependent activation, potentially through localization to specific phosphoinositide-containing organelles by protein-binding partners.

Active Rab11 mediates its roles in membrane trafficking through recruiting a number of protein-binding partners, including motor

protein complexes, and the exocyst complex[53]. Rab11 is also unique among Rab GTPases in that it can interact with the protein-binding partners PI4KB[54] and Rabin8[55] through a unique interface directly over the nucleotide-binding pocket, which allows for the formation of tertiary complexes with Rab11 effectors. PI4KB can bind GDP-bound Rab11[38], suggesting there might be a role in GEF presentation. However, the structure of Rab11 bound to SH3BP5 precludes formation of a ternary complex with PI4KB.

The structure of SH3BP5 bound to Rab11 reveals the mechanism for how SH3BP5 binding leads to nucleotide

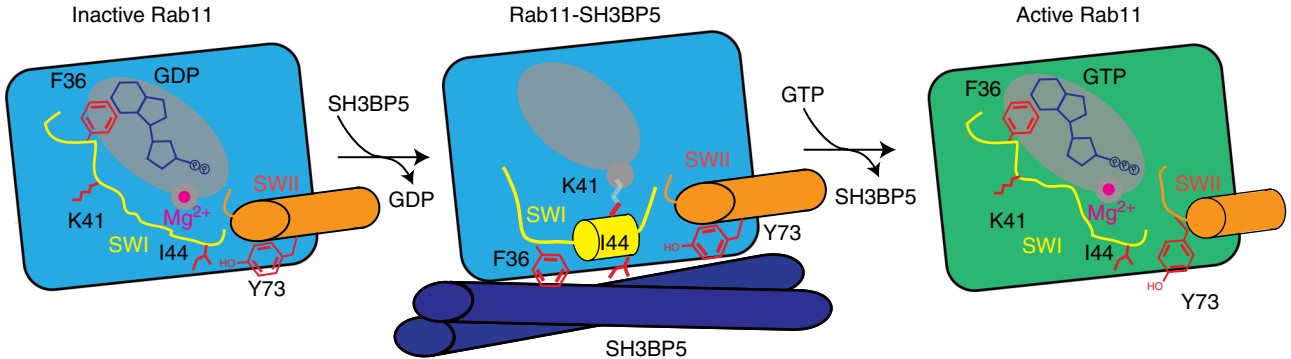

**Fig. 8** Model of Rab11 activation by SH3BP5. In the GDP-bound state switch I (SWI) directly interacts with bound nucleotide, with F36 having a key role in stabilizing this interaction. Upon SH3BP5 binding, switch I undergoes a large conformational rearrangement, with F36 and I44 forming hydrophobic contacts with SH3BP5, leading to exposure of the nucleotide-binding pocket. F36 and I44 are in a highly constrained orientation and require the interspersing residues to allow for formation of this interface. Similar to the Ypt7 GEF Mon1-ccz1[44] the K41 residue projects into the $Mg^{2+}$-binding pocket. The lack of clear electron density around K41 is indicated by being colored in gray, to indicate ambiguity in its exact positioning. There are limited conformational changes in switch II upon SH3BP5 binding, with the hydroxyl from Y73 in position to cap the switch I helix. Upon SH3BP5 disengagement, $Mg^{2+}$ and GTP can enter the nucleotide-binding pocket, with active Rab11 able to bind and recruit downstream effectors

exchange and Rab11 specificity (Fig. 8). Switch I residues have a key role in stabilizing bund nucleotide, with F36 and L38 forming a hydrophobic cap over the guanine ring in both GDP- and GTP-bound structures[41]. In the SH3BP5-bound complex there is a substantial conformational change in switch I of Rab11 compared with nucleotide-bound states, which leads to disruption of switch I nucleotide contacts and allows for release of bound nucleotide. Specifically, F36 and L38 undergo an extensive rearrangement to interact with SH3BP5 and this disrupts their ability to interact with bound nucleotide. Mutation of either F36 or L38 to alanine surprisingly does not lead to a large increase in the intrinsic exchange rate, as seen for the corresponding F33 in Ypt7 (equivalent to F36 in Rab11)[44]; however, mutation of F36 to alanine prevents SH3BP5-mediated nucleotide exchange. The disruption of the interaction of the aromatic residue located in SwI with nucleotide has a key role in nucleotide exchange in TRAPP[52] and DENND1[43], and this is conserved in SH3BP5. The large conformational change in switch I is characteristic of previously solved Rab-GEF structures[42–44,46,47,52,56]. The SH3BP5 coiled coil that interacts with Rab11 shares the closest structural homology to the coiled coil archaeal chaperone prefoldin[39], and this ability to stabilize partially unfolded proteins reveals a potential evolutionary path to stabilizing the partially unfolded Rab11 nucleotide-free state.

SH3BP5 is highly selective for Rab11 and this is likely driven through a highly constrained conformation of the hydrophobic residues F36 and I44 in switch I. This makes Rab11 extremely sensitive to mutation of switch I residues between F36 and I44, which prevent this constrained F36-I44 orientation. The K41 in switch I projects into the $Mg^{2+}$-binding pocket, similar to seen in the Ypt7-Mon1-Ccz1 complex[44]. However, mutation of this residue in Rab11 leads to only a minor decrease in GEF efficiency, with this suggesting it does not play as critical a role as for Ypt7-Mon1-Ccz1. Rab11A and Rab11B are mutated in developmental disorders[16,17], leading to encephalopathies and co-occurrence of seizures and intellectual disability. The V22M, S154L, and K24R all destabilized Rab11, either through a complete (V22M, S154L) or partial (K24R) disruption of nucleotide binding. K13N almost completely prevented SH3BP5-mediated nucleotide exchange, with R82C and A68T both leading to slightly decreased rates. This reveals that clinically relevant Rab11 mutations disrupt SH3BP5-mediated exchange.

The use of a genetically encoded Rab11 FRET sensor[33] revealed that transfection of SH3BP5 leads to a significant increase in

Rab11 activation, with no increase seen with SH3BP5 GEF-deficient mutations. Fluorescently labeled SH3BP5 and AS-Rab11 colocalize, with SH3BP5 primarily localizing to intracellular membranes. Among the key unanswered questions are the mechanisms by which SH3BP5 is localized to specific intracellular membranes and whether SH3BP5 is required for activation of a specific Rab11 pool. As SH3BP5 has key roles in regulation of the signaling kinases BTK and JNK through a non-GEF-mediated process, the molecular details presented here provide an approach to define the roles of SH3BP5 in Rab11-dependent processes. Misregulation of Rab11 isoforms is a key driver in oncogenesis (Rab25) and developmental diseases (Rab11A and Rab11B), and our work reveals insight into how Rab11 can be activated. Moreover, our results not only reveal the specifics of Rab11-GEF activation but also provide important insight into the mechanisms by which Rab GEFs achieve specificity.

## Methods

**Plasmids and antibodies.** The full-length SH3BP5 gene was provided by MGC (DanaFarber HsCD00326538 [https://plasmid.med.harvard.edu/PLASMID/GetCloneDetail.do?cloneid = 326538&species = ]). pDONR223-SH3BP5(31-455) was a gift from William Hahn and David Root (Addgene plasmid #23579 [https://www.addgene.org/23579/]). Rab25 (HsCD00327861 [https://plasmid.med.harvard.edu/PLASMID/GetCloneDetail.do?cloneid = 327861&species = ]), SH3BP5L (HsCD00323009 [https://plasmid.med.harvard.edu/PLASMID/GetCloneDetail.do?cloneid = 323009&species = ]), Rab8a (HsCD00044586 [https://plasmid.med.harvard.edu/PLASMID/GetCloneDetail.do?cloneid = 44586&species = ]), Rab4b (HsCD00296539 [https://plasmid.med.harvard.edu/PLASMID/GetCloneDetail.do?cloneid = 296539&species = ]), Rab2a (HsCD00383517 [https://plasmid.med.harvard.edu/PLASMID/GetCloneDetail.do?cloneid = 383517&species = ]), Rab14 (HsCD00322387 [https://plasmid.med.harvard.edu/PLASMID/GetCloneDetail.do?cloneid = 322387&species = ]), and Rab12 (HsCD00297182 [https://plasmid.med.harvard.edu/PLASMID/GetCloneDetail.do?cloneid = 297182&species = ]) were purchased from the Dana Farber Plasmid Repository. Genes were subcloned into vectors containing a N-terminal GST-tag and, in some cases, a C-terminal His-tag by Gibson assembly. Rab11 and SH3BP5 substitution mutations were generated using site-directed mutagenesis according to published protocols, and C-terminal and N-terminal residues of SH3BP5 were removed using Gibson ligation-independent assembly[57]. The following antibodies were used in this study: rabbit anti-SH3BP5 (SIGMA, HPA057988, IF 1:50) and anti-rabbit IgG Alexa Fluor 568 (ThermoFisher, A-11036, IF 1:1000).

**Bioinformatics.** Sequences were aligned using Clustal Omega Multiple Sequence Alignment and the aligned sequences were subsequently analyzed by ESPript 3.0 to visualize conserved regions. The protein interaction interfaces from the asymmetric unit was examined using the PDBePISA (Proteins, Interfaces, Structures and Assemblies) server[58]. The SH3BP5 structure was compared with similar PDB structures using the DALI server[59]. The surface potential map was generated using the APBS server[60].

**Protein expression.** Constructs of SH3BP5 and Rab11 were all expressed in BL21 C41 *Escherichia coli*. Rab11 was induced with 0.5 mM IPTG (isopropyl β-D-1-thiogalactopyranoside) and grown at 37 °C for 4 h. Rab25 was expressed in Rosetta cells, induced with 0.1 mM IPTG and grown overnight at 23 °C. SH3BP5 and the remaining Rab proteins were induced with 0.1 mM IPTG and grown overnight at 23 °C. SeMet Rab11 and SH3BP5 were expressed in B834 *E. coli* in minimal media with SeMet (Molecular Dimensions), induced with 0.1 mM IPTG, and grown overnight at 23 °C. Pellets were washed with ice-cold phosphate-buffered saline, flash frozen in liquid nitrogen, and stored at − 80 °C until use.

**Protein purification.** Cell pellets were lysed by sonication for 5 min in lysis buffer (20 mM Tris pH 8.0, 100 mM NaCl, 2 mM β-mercaptoethanol (BME), and protease inhibitors (Millipore Protease Inhibitor Cocktail Set III, Animal-Free)). Triton X-100 was added to 0.1% v/v and the solution was centrifuged for 45 min at 20,000 × $g$ at 1 °C. Supernatant was loaded onto a 5 ml GSTrap 4B column (GE) in a superloop for 2 h and the column was washed in Buffer A (20 mM Tris pH 8.0, 100 mM NaCl, 2 mM BME) to remove nonspecifically bound proteins. The GST-tag was cleaved by adding Buffer A containing 10 mM BME and TEV protease to the column and incubating overnight at 4 °C. Cleaved protein was eluted with Buffer A. Protein was further purified by separating on a 5 ml HiTrap Q column with a gradient of Buffer A and Buffer B (20 mM Tris pH 8.0, 1 M NaCl, 2 mM BME). Protein was pooled and concentrated using Amicon 30 K concentrator and size exclusion chromatography (SEC) was performed using a Superdex 200 increase 10/300 or Superdex 75 10/300 column equilibrated in SEC Buffer (20 mM HEPES pH 7.5, 500 mM NaCl, 0.5 mM TCEP). Rab proteins not used for crystallography were purified using SEC Buffer 2 (20 mM HEPES pH 7.5, 150 mM NaCl, 1 mM MgCl$_2$, and 0.5 mM TCEP). Fractions containing protein of interest were pooled, concentrated, flash frozen in liquid nitrogen, and stored at − 80 °C.

Protein for crystallization was generated through mixing the SH3BP5 truncations and Rab11(Q70L) described above at an equimolar amount. The protein mixture was incubated for 5 min at 4 °C. EDTA (pH 7.5) was added to a final concentration of 20 mM and the solution was left to incubate for 1 h at 4 °C. The protein complex was centrifuged for 3 min at 21,130 × $g$ and loaded onto a Superdex 200 10/300 column to separate the complex from free nucleotide. Fractions containing protein of interest were pooled, concentrated, flash frozen in liquid nitrogen and stored at − 80 °C.

**Lipid vesicle preparation.** Nickelated lipid vesicles were made to mimic the composition of the Golgi organelle [15% phosphatidylethanolamine (egg yolk PE, Sigma), 20% phosphatidylinositol (soybean PI from Avanti), 10% phosphatidylserine (bovine brain PS, Sigma), 45% phosphatidylcholine (egg yolk PC Sigma), and 10% DGS-NTA(Ni) (18:1 DGSNTA(Ni), Avanti)]. All other vesicle compositions are described in Supplementary Figure 2. PI3P and PI4P were obtained from Avanti. Vesicles were prepared by combining liquid chloroform stocks together at appropriate concentrations and evaporating away the chloroform with nitrogen gas. The resulting lipid film layer was desiccated for 20 min before being resuspended in lipid buffer (20 mM HEPES (pH 7.5) and 100 mM KCl) to a concentration of 2.0 mg/mL or 1 mg/ml. The lipid solution was vortexed for 5 min, bath sonicated for 10 min, and flash frozen in liquid nitrogen. Vesicles were then subjected to three freeze thaw cycles using a warm water bath. Vesicles were extruded 11 times through a 100 nm NanoSizer Liposome Extruder (T&T Scientific) or a 400 nm NanoSizer Liposome Extruder (T&T Scientific) and stored at − 80 °C.

**In-vitro GEF assay.** C-terminally His-tagged Rab11 was purified as described above. Rab11 was preloaded for the assay by adding EDTA to a final concentration of 5 mM and incubating for 30 mins prior to adding fivefold excess of Mant-GDP (ThermoFisher Scientific). Magnesium chloride was added to 10 mM to terminate the loading process and the solution was incubated for 30 min at 25 °C. SEC was performed using a Superdex 75 10/300 column in SEC Buffer 2 (20 mM HEPES pH 7.5, 150 mM NaCl, and 1 mM MgCl$_2$, 0.5 mM TCEP) to remove any unbound nucleotide. Fractions containing Mant-GDP loaded Rab11 were pooled, concentrated, flash frozen in liquid nitrogen, and stored at − 80 °C. Reactions were conducted in 10 μl volumes with a final concentration of 4 μM Mant-GDP loaded Rab11, 100uM GTPγS, and SH3BP5(9nM-1μM) or SH3BP5L. Rab11 and membrane (0.2–0.4 mg/ml, see Supplementary Figure 2) were aliquoted into a 384-well, black, low-volume plate (Corning 3676). To start the reaction, SH3BP5 and GTPγS were added simultaneously to the wells and a SpectraMax® M5 Multi-Mode Microplate Reader was used to measure the fluorescent signal for 1 h (Excitation λ = 366 nm; Emission λ = 443 nm). Data was analyzed using GraphPad Prism 7 Software, and $k_{cat}/K_m$ analysis was carried out according to the protocol of ref. [61]. In brief, GEF curves were fit to a non-linear dissociate one phase exponential decay using the formula $I(t) = (I_0 − I_∞)*\exp(− k_{obs}*) + I_∞$ (GraphPad Software), where $I$ ($t$) is the emission intensity as a function of time, and $I_0$ and $I_∞$ are the emission intensities at $t$ = o and $t$ = ∞. The catalytic efficiency $k_{cat}/K_m$ was obtained by a slope of a linear least squares fit to $k_{obs} = k_{cat}/K_m*[GEF] + k_{intr}$, where $k_{intr}$ is the rate constant in the absence of GEF.

**Crystallography.** Crystallization trials of the SH3BP5-Rab11 complex were set using a Crystal Gryphon liquid handling robot (Art Robbins Instruments) in 96-well Intelli-Plates using sitting drops at 18 °C. Initial hits were obtained in the Index HT crystallization kit (Hampton Research) and refinement plates for Index HT condition F11 (25% (w/v) PEG-3350, 200 mM sodium chloride, 100 mM Bis-Tris pH 6.5) were set. The best crystals of the optimized SH3BP5 construct with Rab11A were obtained in 2 μL hanging drops, with a reservoir solution of 23% (w/v) PEG-3350, 200 mM sodium chloride, 100 mM Bis-Tris pH 6.5, with a ratio of 4:1 protein/reservoir. Crystals were frozen in liquid nitrogen using cryo buffer [24% PEG-3350 (w/v), 200 mM sodium chloride, 100 mM Bis-Tris pH 6.5, 15% (v/v) ethylene glycol]. SeMet containing crystals were obtained in 1.6 μL hanging drops with a reservoir solution of 14% (w/v) PEG-3350, 200 mM NaBr, 100 mM Bis-Tris pH 5.5, and 5% Tacsimate pH 6.0 at a ratio of 4:1 protein to reservoir. Crystals were frozen in liquid nitrogen using cryo buffer 2 [20% (w/v) PEG-3350, 150 mM NaCl, 50 mM NaBr, 100 mM Bis-Tris pH 5.5, 5% Tacsimate pH 6.0, and 15% (v/v) ethylene glycol].

Diffraction data were collected at 100 K at beamline 08ID-1 of the Canadian Macromolecular Crystallography Facility (Canadian Light Source). Native data were collected at a wavelength of 0.97949 Å and SeMet data was collected at 0.97895 Å. Data were integrated using XDS[62] and scaled with AIMLESS[63]. Datasets were collected for both native SH3BP5 (1–265)/Rab11 (1–213) and SeMet incorporated SH3BP5 (1–265)/Rab11 (1–213). Full crystal collection details are shown in Table 1. Initial phases were determined by single-wavelength anomalous dispersion at the selenium peak energy with initial phases, density modification and automated model building carried out using CRANK (version 2.0)[64]. This allowed for an initial model of SH3BP5 to be build and the location of Rab11 was identified in the asymmetric unit through molecular replacement using PHASER[65], with the structure of GDP-bound Rab11 used as the search model with both switch I and switch II removed[41]. The final model of SH3BP5-Rab11 complex was built using iterative model building, including manual rebuilding of the Rab11 switches in COOT[66] and refinement using phenix.refine[67] to a $R_{work}$ = 23.54 and $R_{free}$ = 27.80. Ramachandran statistics for final model—favored 96.6%, outliers 0.51%. Full crystallographic statistics are shown in Table 1.

**Identification of disordered regions in SH3BP5 using HDX-MS.** HDX reactions were conducted in 50 μl reaction volumes with a final concentration of 0.2 μM SH3BP5 (1–455) per sample. Exchange was carried out in triplicate for a single time point (3 s at 1 °C) and all steps were carried out in a 4 °C cold room. Hydrogen deuterium exchange was initiated by the addition of 48 μl of D$_2$O buffer solution (10 mM HEPES pH 7.5, 50 mM NaCl, 97% D$_2$O) to the protein solution, to give a final concentration of 93% D$_2$O. Exchange was terminated by the addition of acidic quench buffer at a final concentration 0.6 M guanidine-HCl and 0.9% formic acid. Samples were immediately frozen in liquid nitrogen at -80 °C.

**Mapping changes in Rab11 mutants using HDX-MS.** All clinically relevant Rab11 mutants were purified identically to WT Rab11. HDX reactions were conducted in 50 μl reaction volumes with a final concentration of 0.5 μM Rab11(WT, V22M, K24R, or S154L) per sample. Exchange was carried out in triplicate for two time points: 3 s at 1 °C and 300 s at 18 °C. Hydrogen deuterium exchange was initiated by the addition of 49 μl of D$_2$O buffer solution (10 mM HEPES (pH 7.5), 50 mM NaCl, 97% D$_2$O) to the protein solution, to give a final concentration of 95% D$_2$O. Exchange was terminated by the addition of acidic quench buffer at a final concentration 0.6 M guanidine-HCl and 0.9% formic acid. Samples were immediately frozen in liquid nitrogen at − 80 °C.

**Mapping of the SH3BP5-Rab11-binding interface using HDX-MS.** HDX reactions were conducted in 20 μl reaction volumes with a final concentration of 1.0 μM Rab11(Q70L) and 1.0 μM SH3BP5 (31–455) per sample. Exchange was carried out in triplicate for four time points (3 s at 1 °C and 3 s, 30 s and 300 s at room temperature). Before the addition of D$_2$O, both proteins were incubated on ice in the presence of 20 μM EDTA for 1 h to facilitate release of nucleotide. Hydrogen deuterium exchange was initiated by the addition of 17.5 μl of D$_2$O buffer solution (10 mM HEPES pH 7.5, 500 mM NaCl, 97% D$_2$O) to 2.5 μl of the protein solutions, to give a final concentration of 78% D$_2$O. Exchange was terminated by the addition of acidic quench buffer at a final concentration 0.6 M guanidine-HCl and 0.9% formic acid. Samples were immediately frozen in liquid nitrogen at − 80 °C.

**Investigating the role of membrane using HDX-MS.** HDX reactions were conducted in 20 μl reaction volumes with a final concentration of 0.4 μM C-terminally His-tagged Rab11A (1–211), 0.4 μM SH3BP5 (31–455), and 0.2 mg/mL nickelated lipid vesicles [15% PE, 20% PI, 10% PS, 45% PC, and 10% DGS-NTA(Ni)] per sample. Exchange was carried out in triplicate for four time points (3 s, 30 s, 300 s, 3000 s) at room temperature. Before the addition of D$_2$O, 1 μl of 20 μM Rab11 and 2 μl of 2 mg/mL membrane (or membrane buffer) were left to incubate for 30 s. One microliter of 20 μM SH3BP5 was then added and incubated a further 30 s prior to the initiation of hydrogen deuterium exchange by the addition of 16 μl of D$_2$O buffer solution (10 mM HEPES pH 7.5, 200 mM NaCl, 97% D$_2$O) to the samples to give a final concentration of 77% D$_2$O. Exchange was terminated by the addition of

acidic quench buffer giving a final concentration 0.6 M guanidine-HCl and 0.9% formic acid. Samples were immediately frozen in liquid nitrogen at -80 °C.

**Mutational analysis of SH3BP5 using HDX-MS.** HDX reactions were conducted in 50 μl reaction volumes with a final concentration of 0.6 μM SH3BP5(WT) or 0.6 μM SH3BP5(LNQ52AAA), or 0.6 μM SH3BP5(LE250AK) per sample. Exchange was carried out in triplicate for two time points (3 s, 300 s at 18 °C). Hydrogen deuterium exchange was initiated by the addition of 48.5 μl of $D_2O$ buffer solution (10 mM HEPES pH 7.5, 200 mM NaCl, 97% $D_2O$) to the samples to give a final concentration of 94% $D_2O$. Exchange was terminated by the addition of acidic quench buffer giving a final concentration 0.6 M guanidine-HCl and 0.9% formic acid. Samples were immediately frozen in liquid nitrogen at − 80 °C.

**HDX-MS data analysis.** Protein samples were rapidly thawed and injected onto an ultra-performance liquid chromatography (UPLC) system kept in a cold box at 2 °C. The protein was run over two immobilized pepsin columns (Applied Biosystems; Porosyme 2-3131-00) and the peptides were collected onto a VanGuard Precolumn trap (Waters). The trap was eluted in line with an ACQUITY 1.7 μm particle, $100 \times 1$ mm$^2$ C18 UPLC column (Waters), using a gradient of 5%-36% B (Buffer A 0.1% formic acid, Buffer B 100% acetonitrile) over 16 min. MS experiments were performed on an Impact QTOF (Bruker) and peptide identification was done by running tandem MS (MS/MS) experiments run in data-dependent acquisition mode. The resulting MS/MS datasets were analyzed using PEAKS7 (PEAKS) and a false discovery rate was set at 1% using a database of purified proteins and known contaminants. HDExaminer Software (Sierra Analytics) was used to automatically calculate the level of deuterium incorporation into each peptide. All peptides were manually inspected for correct charge state and presence of overlapping peptides. Deuteration levels were calculated using the centroid of the experimental isotope clusters. Fully deuterated samples were generated by incubating SH3BP5 with 3 M guanidine for 30 min before the addition of $D_2O$. The protein was exchanged for 1 h on ice before adding quench buffer. This fully deuterated sample allows for the control of peptide back exchange levels during digestion and separation. Differences in exchange were in a peptide were considered significant if they met all three of the following criteria: > 5% change in exchange, > 0.5 Da difference in exchange, and a $p$-value < 0.01 using a two-tailed Student's $t$-test. Samples were only compared within a single experiment and were never compared with experiments completed at a different time with a different final $D_2O$ level.

**Rab11 activated sensor cellular experiments.** HEK 293 T cells ($2 \times 10^5$; ATCC, CRL-11268) were plated in a six-well plate and Lipofectamine 2000 (Invitrogen) was used for transfection. Less than 500 ng of DNA were transfected in every condition. After 36 h of transfection, lysis was performed in lysis buffer (50 mM Tris-HCl, pH 7.4, 1% Tritorn X-100, 10 mM $MgCl_2$, 100 mM NaCl, proteinase inhibitors) and lysate was measured in a fluorometer cuvette. The Fluoromax-4 Horiba fluorometer was used to perform the measure. Laser excitation at 433 nm was used and the emission spectrum from 450 to 550 nm was recorded. A second measurement was made by directly exciting YFP at 505 nm and measuring its emission at 525 nm, to normalize for biosensor concentration. Colocalization analysis was performed using ImageJ JACOP plugin. Pearson's coefficient of correlation was calculated using Costes' automatic threshold.

**Quantification of Rab11 activity in COS-7 cells.** The sensitized FRET and CFP images acquired from transfected COS-7 cells (ATCC, CRL-165), were smoothed using Gaussian blur and background subtraction was performed according to previous published protocol[68]. Afterwards, FRET activity ratio was computed by dividing sensitized FRET pixels by the CFP pixels, excluding saturated signals.

**Statistical analysis.** Six independent experiments ($n$) were performed for microscopy-based experiments, and statistical significance were obtained. Means ± SEM were used to present values. $P$-values were calculated using one-way analysis of variance followed by Bonferroni's multiple comparison post test (GraphPad Software). The following legends are used for statistical significance: *$P < 0.05$, **$P < 0.01$, and ***$P < 0.005$.

For all GEF and HDX-MS assays, experiments were carried out in triplicate and means ± SD are shown in all figures. Statistical analysis between conditions was performed using a two-tailed Student's $t$-test, with $p$-values shown the same as described for cellular experiments.

## Data availability

The structure factors and coordinates for the structure of Rab11A bound to SH3BP5 have been deposited in the protein databank with the accession code 6DJL. The mass spectrometry proteomics data have been deposited to the ProteomeXchange Consortium via the PRIDE[69] partner repository with the dataset identifier PXD010586. The processed HDX-MS data are provided as Supplementary Data 1. All other data supporting the findings of this study are available from the corresponding author on reasonable request.

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

## Acknowledgements

J.E.B. is supported by a new investigator grant from CIHR, a discovery research grant from the Natural Sciences and Engineering Research Council of Canada (NSERC-2014-05218), and a Michael Smith Foundation for Health Research (MSFHR) Scholar award (17686). M.J.B. was supported by Canadian Institutes of Health Research Grant 148596. M.J.B. gratefully acknowledges the Canada Research Chair program for salary support. J. P.M. and E.H. are supported by AIRC and Worldwide Cancer Research grants. We acknowledge the staff at the Canadian Light Source (CLS) and the Stanford Synchrotron Radiation Lightsource (SSRL). The CLS is supported by the Natural Sciences and Engineering Research Council of Canada, the National Research Council Canada, the Canadian Institutes of Health Research, the Province of Saskatchewan, Western Economic Diversification Canada, and the University of Saskatchewan.

## Author contributions

J.E.B. and M.L.J. designed all biophysical/biochemical experiments. M.L.J., J.T.B.S., and R.H. carried out protein expression/purification. M.L.J. and J.T.B.S. carried out all biochemical studies. J.E.B., M.L.J., J.T.B.S., and J.A.M. performed crystallographic work. J.E. B. and M.J.B. carried out crystallographic data analysis. M.L.J., D.J.H., and J.T.B.S. carried out HDX-MS experiments. J.P.M. and E.H. designed and carried out cellular Rab11 activation assays. M.L.J. and J.E.B. wrote the manuscript, with input from all authors.

## Additional information

**Competing interests:** The authors declare no competing interests

