## [Peer Review File · Nature Communications]

Reviewers' comments:

Reviewer #1 (Remarks to the Author):

The paper by Jenkins et al. describes a unique molecular insight into Rab11 regulation and presents a novel crystal structure of SH3BP5 bound to Rab11. Overall the paper is well written and the results are mostly consistent with the data presented in the paper. However, unless my comments are addressed I cannot recommend publication of this manuscript.

Specific comments:

1. First of all I didn't understand, nor it was explained in the text why the mutant Q70L Rab11A was used throughout the study and not the WT? In Figure S7 in the FRET experiments the authors used WT Rab11 but nowhere else in the manuscript was WT used. Can you please explain?
2. Page 11, line 225, the authors claim that V22M and S154L mutants are "expected to dramatically disrupt binding"; how was this hypothesis tested? Is there data to support this??
3. Page 12 line 264; 265. The authors said that those triple and double mutants were tested with HDX to ensure that the protein was properly folded. There is no HDX data in current manuscript that looks at the LNQ52AAA and LE250AK. If this data exist I would like to see it.
4. Page 15, line 326. The authors say that residues F36 and I44 are crucial for the SH3BP5 interactions. Were these two residues mutated and then the new constructs tested for binding? I would like to see mutant data to support this hypothesis. Also, the numbering that the authors used has to be revised: F36 is actually F39 in the peptide map data and I44 is actually I47 in the peptide data; so I would recommend that the authors revise their sequence numbering system otherwise its very confusing.
5. Page 21, line 467; the authors claim that they measured exchange at 0.3sec. Can you please explain how was this achieved? What kind of device was used to measure such short labeling times?
6. For the labeling experiments why were there different D2O dilutions between the experiments; that are obviously leading to different % of D2O being incorporated. How was the data interpreted? Was there a general correction done or not? Otherwise the results are not well interpreted if data from one labeling experiment was compared with data from another labeling experiment that used a lower % of D2O. Please explain.
7. Page 22, line 486; for the complex and membrane bound experiments they used a different Rab 11 construct compared with the previous HDX experiments: Rab11(Q70L) versus Rab11A (1-211). Why is that? Please explain.
8. Page 23, line 511: how do the authors know that they obtained a fully deuterated protein sample? How was that measured? Also, what was the level of back exchange throughout the experimental setup and what was the error of the measurements? The parameters that they used to define significant differences seem a bit loose; >0.5% and >0.5 Da in exchange. 0.5% in a short peptide (less than 10 amino acids) is very different compared to a 0.5% in a longer peptide (15-20 amino acids).
9. Supplemental fig S3: line 930; was deuteration performed at 0.3 sec or 3 sec? In the experimental section the authors say that deuteration was done at 0.3sec to identify disordered regions; please explain.
10. Figure S3, panel E: for forming the complex the authors used SH3BP5 (1-265); and then for the HDX exp with the complex they used the SH3BP (31-455) construct. Why is that? Do you also have HDX data for the 1-265 alone and in Rab11 complex? Otherwise can you please explain? Also, looking at the SEC trace is seems that 1-265 is a dimer in solution. Is this true? You never mention that in the text.
11. Figure S5: panel A; the peptides shown on the crystal structure are not clearly marked; therefore its confusing to understand what is represented.

Reviewer #2 (Remarks to the Author):

This manuscript by Jenkins et al. presents the biochemical and structural characterization of the Rab11 GEF SH3BP5, including a crystal structure of a catalytic complex, GEF activity assays and cellular studies with a FRET activation sensor. Based on the same basic principle, SH3BP5 employs yet a novel mechanism to promote nucleotide exchange, adding to the divers family of RabGEFs. SH3BP5 forms a v-shaped coil-coil structure, with one leg binding to Rab11. Switch I is distorted and bound in a unique confirmation, and a Switch I Lys residue is inserted into the nucleotide-

binding pocket similar to the Mon1-Ccz1/Rab7 exchange mechanism. The relevance of both mechanisms is confirmed by mutational analysis. Activity of SH3BP5 was stimulated when GEF assays were performed on membranes, though not as dramatically as for some other GEFs. Finally, GEF activity deficient mutants were also analyzed in cell culture experiments with a FRET-based Rab11 activation sensor.

This work adds to our understanding of Rab activation. However, several aspects need to be addressed before publication.

1. The SH3BP5 membrane interactions should be addressed in more detail. First, a surface potential map of SH3BP5 would be helpful to judge if an interaction with PIPs is plausible. Or does the unstructured C-terminal tail contain basic stretches? Second, are there amphipathic helices that might sense packing defects? The stimulation of GEF activity in the presence of membrane suggests that some kind of protein lipid interaction must exist.

Previous studies suggest that PIPs play an important role in Rab11 activation, but the authors did not observe any effect of PIPs on GEF activity in the presence of liposomes. With 100 nm diameter and 20 % PE, liposomes contain packing defects that could lead to membrane association so that an effect of PIPs might be masked. Liposomes with low DGS NTA, no PE and larger diameter should be tested as well.

2. The authors depict inactive Rab11 GDP as Mg free structure, which is very unlikely. There was a crystal structure reported of Mg-free Rab11 GDP, but this structure shows a dimeric Rab11 with the switch I regions involved in dimerization. This crystallization artefact probably led to GDP binding without Mg, but such a conformation does not seem plausible in a physiological context. Fig. 7 should be changed and the discussion revised.

3. For the cellular activation assay, FRET signals were quantified spectroscopically from cell lysates (Fig 6F). Although effects can be observed, they seem rather weak compared to the changes in FRET in the cell images (Fig 6D). It would make more sense to quantify the cellular FRET? Figure 6D should contain panels showing Rab11 localization and the data for an experiment with AS-Rab11 alone. Does Rab11 localization change? To what extent do Rab11 activation and SH3BP5 localization correlate? Are the SH3BP5 mutants localized like wild-type?

4. The analysis of Rab11 patient mutations does not add much to the manuscript, although the assumptions are all reasonable. At least the point mutations that are suspected to effect GEF activity should be tested in vitro to substantiate this part (K13N, R82C, and A68T as a control).

Additional points:

- SH3BP5 runs at an apparent MW of 78 kDa on gel filtration. Is this the result of dimerization or the v-shape? Could any of the crystallographic contacts represent a dimerization interface in solution. The references to Fig S3 are wrong in the text (l. 155 and 157)
- For the Fo-Fc omit map, side chains should be shown as well – it would be nice to see that K41 is well defined.
- Axis units in Fig 1C and 5B should be 10E-3
- All shown side chains should be labeled in Fig 4A, and the font size should be increased.
- The comparison of Sec2 and SH3BP5 does not suggest convergent evolution, but simply independent evolution: the two GEFs mediate nucleotide exchange by different interactions/interfaces, they just happen to both be coiled-coils.
- A wt reference curve would be helpful in the panels of Fig 5A.
- Not every data point in Fig 5B has an error bar, thus only represents a single experiment?

Reviewer #3 (Remarks to the Author):

The manuscript from the Burke group describes the previously unknown crystal structure of the small GTPase in complex with its cellular activator SH3BP5.

Small GTPases control crucial signaling events in the eukaryotic cell. In order to become active, they require guanine nucleotide exchange factors (GEFs) that replace the tightly bound GDP-nucleotide with GTP. Until now, the mechanism of activation of the trafficking regulator GTPase Rab11 was unknown. The authors therefore have characterized the complex of Rab11 with its GEF SH3BP5 structurally and biochemically. For the first time, the mechanism of Rab11-activation is understood. Thus, these data may be of use when analyzing Rab11 dependent intracellular transport processes.

The manuscript is well written and the data are of the highest quality. There is no doubt on the technical excellence of the study. I therefore only have a general point of criticism (see below),

but I recommend the article to Nature Communications after proper amendment.

Major comments

- page 11, from 243: The authors aim at analyzing the determinants for SH3BP5-specificity towards Rab11. To this purpose, they substitute Rab11 positions being apparently crucial for GEF-binding with the corresponding amino acids from other Rabs (L38P, S40F, K41P). However, the mutations they introduce by the approach may affect the structure of Rab11 severely, both locally and globally. This argument applies to the proline substitutions especially, since they may alter the proper conformations and positioning of the adjacent regions and may also introduce or abrogate crosstalk with other side chains. I therefore consider this approach as being inappropriate to understanding the importance of individual Rab11 amino acids or for confirming the Rab11:SH3BP5-complex interface on the basis of the crystal structure.

Rather, the author should alter positions L38, S40, K41, and others into alanine and quantify the effect of GEF-mediated exchange. Alternatively, they may create SH3BP5 alanine mutants for the same purpose. Confirming the complex interface as observed in the crystal structure should be included.

However, I believe that the authors have correctly interpreted their structural data. This final biochemical analysis is simply required as an ultimate confirmation.

Minor comments

- General comment: I recommend that the authors add a (sub-)section describing the GEF-reaction. How is the nucleotide believed to be displaced from its binding pocket? From the manuscript I conclude that F36 should have a major impact. If this is the case, the authors should clearly mention the role of F36 in binding to the guanine base of GDP/GTP and its relevance for high nucleotide affinity. I could not find such a statement or a similar one in the main text of the manuscript.

- page 10, from line 220: The authors speculate on the potential consequences of disease-related Rab11 amino acid substitutions for the interaction with SH3BP5. However, not data are presented and thus this part remains rather speculative. It is advised to move the section into the discussion.

- I recommend including a figure/supplementary figure that demonstrates the most important amino acid interactions of the Rab11-SH3BP5 interface (in particular F36 and I41 of Rab11). This will facilitate understanding the crucial involvement of the respective side chains for the complex formation.

- author statement "Intriguingly, dominant negative mutants of 135 Rab11A (S25N) caused rapid release of mant-GDP even in the absence of SH3B": It is well known that the mutation affects Mg²⁺ coordination and therefore decreases nucleotide affinity (i.e. increases dissociation rate of the nucleotide). Since the finding is not surprising, please amend accordingly (or delete the sentence since the finding is of no added value for the manuscript).

- the supplementary movie is a good visual aid to the reader. Nevertheless, the authors should try to improve

a) the definition of the helices. In particular, the N- and C-terminal ends of the helices appear to be defined awkwardly. Some regions are depicted as helices but are obviously not.

b) the bond geometry of the side chains. For instance, the phenyl ring of the switch 1 phenylalanine is apparently being distorted during the morph.

Reviewers' comments:

Reviewer #1 (Remarks to the Author):

The paper by Jenkins et al. describes a unique molecular insight into Rab11 regulation and presents a novel crystal structure of SH3BP5 bound to Rab11. Overall the paper is well written and the results are mostly consistent with the data presented in the paper. However, unless my comments are addressed I cannot recommend publication of this manuscript.

We appreciate the overall positive assessment of the manuscript, and we have addressed all of the reviewer's comments as described in the following points.

Specific comments:

1. First of all I didn't understand, nor it was explained in the text why the mutant Q70L Rab11A was used throughout the study and not the WT? In Figure S7 in the FRET experiments the authors used WT Rab11 but nowhere else in the manuscript was WT used. Can you please explain?

We chose the Q70L construct for crystallography studies as there was a slightly increased GEF activity against Q70L vs WT Rab11 (see Fig. S2D,E), indicating that this might form a slightly more stable complex. HDX-MS experiments mapping the SH3BP5 binding interface on Rab11 also used Q70L to verify the crystallographic interface in solution (see Fig. S5A). For all other HDX-MS and biochemical assays they were carried out with WT Rab11 unless indicated clearly in the legend/text. We apologise for not more clearly describing this in the text, and in the revision have clearly described the constructs used and why in more detail (lines 202-204).

2. Page 11, line 225, the authors claim that V22M and S154L mutants are “expected to dramatically disrupt binding”; how was this hypothesis tested? Is there data to support this??

We had originally hypothesized that V22M and S154L would disrupt nucleotide binding based on steric hinderance of nucleotide binding based on the location of these mutations. We have now experimentally verified that V22M and S154L strongly destabilise Rab11 through HDX-MS experiments which reveal greatly increased H/DX through the entire protein, with K24R also partially destabilising Rab11. This data is now included in Fig. S7E,F, and described in text lines 318-322.

3. Page 12 line 264; 265. The authors said that those triple and double mutants were tested with HDX to ensure that the protein was properly folded. There is no HDX data in current manuscript that looks at the LNQ52AAA and LE250AK. If this data exist I would like to see it.

This was an oversight as this data was accidentally omitted from supplemental table I in the first draft. We now show this HDX data in Fig. S7G, with the full raw data included in the revised Supplemental Table I. This has also been described in more detail in the text (lines 391-394).

4. Page 15, line 326. The authors say that residues F36 and I44 are crucial for the SH3BP5 interactions. Were these two residues mutated and then the new constructs tested for binding? I

would like to see mutant data to support this hypothesis. Also, the numbering that the authors used has to be revised: F36 is actually F39 in the peptide map data and I44 is actually I47 in the peptide data; so I would recommend that the authors revise their sequence numbering system otherwise its very confusing.

The suggestion to examine F36 and I44 was an excellent suggestion from the reviewer. We have examined the F36A and I44A mutations and find that they massively decrease SH3BP5 mediated GEF activity (>100 fold decrease). This is now included in Fig. 6. This is described in text (356-359). We have also revised the peptide map numbering in Table S1, this occurred due to the inclusion of residues from the tag/cleavage site, with current numbering corresponding exactly with the SH3BP5 sequence.

5. Page 21, line 467; the authors claim that they measured exchange at 0.3sec. Can you please explain how was this achieved? What kind of device was used to measure such short labeling times?

Hydrogen/deuterium exchange experiments were carried out for 3 seconds at 1 degree Celsius. This is roughly equivalent to 0.3 s of deuterium exchange at room temperature (derived from experiments from Bai et al 1993 and Coales et al 2010). 3 seconds of deuterium incorporation is experimentally possible with sufficient training, and the repeatability is highlighted by the very low standard deviation of the HDX measurement (average less than 1% at this timepoint). This was poorly described in the text, and we have now clearly described exactly how the experiment was carried out in the methods, results (lines 176-179) and figures.

6. For the labeling experiments why were there different D2O dilutions between the experiments; that are obviously leading to different % of D2O being incorporated. How was the data interpreted? Was there a general correction done or not? Otherwise the results are not well interpreted if data from one labeling experiment was compared with data from another labeling experiment that used a lower % of D2O. Please explain.

*For all experiments a back correction was applied according to the % deuterium incorporated (details now fully described in methods). **However, importantly no comparisons were carried out between experiments with different levels of D2O incorporated.***

Two experiments were carried out for the HDX experiment on SH3BP5 binding to Rab11:

- 1. Q70L Rab11 with SH3BP5.*
- 2. SH3BP5 with Rab11 in the presence/absence of membranes.*

The reason we needed different levels of D2O was due to the use of membrane which technically prevented dilution in D2O to the same concentration. The results of these two experiments are now shown in Fig. S5, with the data being separated into two panels to more clearly indicate it represents two independent experiments. This has also been clearly indicated in both the figure legends and methods.

7. Page 22, line 486; for the complex and membrane bound experiments they used a different

Rab 11 construct compared with the previous HDX experiments: Rab11(Q70L) versus Rab11A (1-211). Why is that? Please explain.

The main reason for this is that the HDX experiments mapping the Rab11 binding site on SH3BP5 were carried out with our membrane localised construct (containing a his tag on the c-terminus) allowing for localisation to NiNTA containing membranes. This allows us to examine complex formation on a membrane surface. For the membrane binding experiment, we wanted to minimise any possible complications from mutations and thus we used the WT Rab11 construct. For the experiment on mapping the SH3BP5 binding interface on Rab11 we wanted to verify the contacts seen in the crystal structure, and used the same construct from crystallography (Q70L).

The rationale for the use of the different constructs is now more clearly described in the text (lines 239-258).

8. Page 23, line 511: how do the authors know that they obtained a fully deuterated protein sample? How was that measured? Also, what was the level of back exchange throughout the experimental setup and what was the error of the measurements? The parameters that they used to define significant differences seem a bit loose; >0.5% and >0.5 Da in exchange. 0.5% in a short peptide (less than 10 amino acids) is very different compared to a 0.5% in a longer peptide (15-20 amino acids).

The fully deuterated sample was generated through pre-incubation of the protein with a denaturant containing solution for 30 minutes, followed by incubation with deuterium for 60 minutes. For all peptides the exchange seen was the same or greater than after 3000 s of on exchange without denaturant.

While it is true that it is difficult to know if a full deuterated sample is obtained, what we can say for sure is that the c-terminus of SH3BP5 had almost the exact same level of exchange at 3 s at 1 degree Celsius compared to 30 min of exchange with denaturant present, indicating that these regions fully exchanged within the short deuterium pulse.

*The back exchange for peptides in the fully deuterated sample varied from 10-50%, with an average deuterium incorporation of ~60%. **The average error of all HDX measurements was ~0.8% (SDs of all HDX measurements are included in supplemental table I).***

*We required changes in exchange greater than >5% (not 0.5%) **and** >0.5 Da to define significant, because using only the difference in the percent deuterium or number of deuterons leads to biasing of either long or short peptides. A 5% change in a short peptide will not cross the 0.5 Da threshold, and a 0.5 Da change in a very long peptide will not cross the 5% threshold. Therefore, requiring changes in **both** is the most appropriate strategy, and along with these differences it must also have a p value < 0.01 based on a two-tailed student t test. This is now more fully described in the methods.*

9. Supplemental fig S3: line 930; was deuteration performed at 0.3 sec or 3 sec? In the experimental section the authors say that deuteration was done at 0.3sec to identify disordered regions; please explain.

Experiments were carried out for 3 s at 1 degree Celsius, as described above. This is now clearly described in the methods and legends.

10. Figure S3, panel E: for forming the complex the authors used SH3BP5 (1-265); and then for the HDX exp with the complex they used the SH3BP (31-455) construct. Why is that? Do you also have HDX data for the 1-265 alone and in Rab11 complex? Otherwise can you please explain? Also, looking at the SEC trace it seems that 1-265 is a dimer in solution. Is this true? You never mention that in the text.

*We have used three different SH3BP5 constructs (FL 1-455, 31-455, and 1-265). These constructs are used in different experiments. **Importantly, all three of these constructs show exactly the same biochemical GEF activity** (new **figure S3D**), suggesting all three bind and interact with Rab11 in a similar fashion. 1-265 is the only construct that formed crystals with Rab11, with HDX experiments using the 31-455 construct.*

We used the 1-265 construct for the gel filtration experiment as we wanted to examine the size/shape of the crystallised complex. For the HDX experiment we used 31-455 to simplify data analysis through removal of peptides from the N-terminus. We would expect the HDX results to be similar to using full length SH3BP5, as the GEF activity between the two is exactly the same, and no clear density was seen from residues 1-38 for SH3BP5 in the crystal structure.

All biochemical assays used the full length SH3BP5. No HDX experiments were carried out on the 1-265 construct, but since it has exactly the same GEF activity, and there was no change in exchange in the region from 266-455 upon Rab11 binding we expect that it would be structurally very similar to the full length SH3BP5.

It is highly unlikely that the SH3BP5 1-265 construct is a dimer in solution, as all intra-SH3BP5 interfaces in the crystallographic unit cell had CSS (complexation significance scores) of 0.0 from the Proteins, Interfaces, Structures and Assemblies (PISA) server. The earlier elution value of SH3BP5 is almost certainly due to its extended v-shaped conformation, and gel filtration primarily is a readout of the shape of a complex. For this reason, we have removed the predicted molecular weights from Fig. S3, as gel filtration is not an accurate measure of MW. Our main point in the figure is that Rab11 and SH3BP5 form a stable complex over gel filtration.

11. Figure S5: panel A; the peptides shown on the crystal structure are not clearly marked; therefore its confusing to understand what is represented.

We have now clearly marked on the structure the identity of the peptides with differences in exchange between conditions.

Reviewer #2 (Remarks to the Author):

This manuscript by Jenkins et al. presents the biochemical and structural characterization of the Rab11 GEF SH3BP5, including a crystal structure of a catalytic complex, GEF activity assays and cellular studies with a FRET activation sensor. Based on the same basic principle, SH3BP5

employs yet a novel mechanism to promote nucleotide exchange, adding to the diverse family of RabGEFs. SH3BP5 forms a v-shaped coil-coil structure, with one leg binding to Rab11. Switch I is distorted and bound in a unique conformation, and a Switch I Lys residue is inserted into the nucleotide-binding pocket similar to the Mon1-Ccz1/Rab7 exchange mechanism. The relevance of both mechanisms is confirmed by mutational analysis. Activity of SH3BP5 was stimulated when GEF assays were performed on membranes, though not as dramatically as for some other GEFs. Finally, GEF activity deficient mutants were also analyzed in cell culture experiments with a FRET-based Rab11 activation sensor.

This work adds to our understanding of Rab activation. However, several aspects need to be addressed before publication.

We appreciate the positive assessment of our manuscript and have addressed all issues as described below.

1. The SH3BP5 membrane interactions should be addressed in more detail. First, a surface potential map of SH3BP5 would be helpful to judge if an interaction with PIPs is plausible. Or does the unstructured C-terminal tail contain basic stretches? Second, are there amphipathic helices that might sense packing defects? The stimulation of GEF activity in the presence of membrane suggests that some kind of protein lipid interaction must exist.

The unstructured c-terminus of SH3BP5 is highly acidic, and as such would not participate in membrane binding. There are no putative amphipathic helices that would be able to bind disordered membranes. Also, HDX experiments with SH3BP5 +/- membranes (no Rab11) showed no differences in exchange, suggesting that if there is membrane interaction it is quite weak. We have added the surface potential map of SH3BP5 in the Fig S4, and have added details on the charge / lack of amphipathic helices, as well as the HDX experiment on SH3BP5 +/- membranes in the text (lines 215-218).

Previous studies suggest that PIPs play an important role in Rab11 activation, but the authors did not observe any effect of PIPs on GEF activity in the presence of liposomes. With 100 nm diameter and 20 % PE, liposomes contain packing defects that could lead to membrane association so that an effect of PIPs might be masked. Liposomes with low DGS NTA, no PE and larger diameter should be tested as well.

We experimentally tested a large set of new membranes. We tested five new membranes (all 400 nm, no PE, with varying levels of PS, PI, PI3P, and PI4P) and found that all membranes stimulated SH3BP5 GEF activity to the same extent as before (~2 fold) (see new Fig. S2). It may be that GEF activity is stimulated solely through the presentation of the Rab11 on a surface, with the only effect being that Rab11 is presented in an orientation conducive to SH3BP5 binding. More discussion on this topic had been added to the text.

2. The authors depict inactive Rab11 GDP as Mg free structure, which is very unlikely. There was a crystal structure reported of Mg-free Rab11 GDP, but this structure shows a dimeric Rab11 with the switch I regions involved in dimerization. This crystallization artefact probably

led to GDP binding without Mg, but such a conformation does not seem plausible in a physiological context. Fig. 7 should be changed and the discussion revised.

We agree with the reviewer that this Mg free structure is most likely a crystallisation artefact due to dimer formation. We have altered Fig. 7 (now Fig. 8 in the revised manuscript) and changed the discussion accordingly.

3. For the cellular activation assay, FRET signals were quantified spectroscopically from cell lysates (Fig 6F). Although effects can be observed, they seem rather weak compared to the changes in FRET in the cell images (Fig 6D). It would make more sense to quantify the cellular FRET? Figure 6D should contain panels showing Rab11 localization and the data for an experiment with AS-Rab11 alone. Does Rab11 localization change? To what extent do Rab11 activation and SH3BP5 localization correlate? Are the SH3BP5 mutants localized like wild-type?

We thank the reviewer for his/her comment. We agree with the reviewer that it is essential to demonstrate that our biosensor is also able to detect changes in Rab11 activity in whole cells, and compare it with spectrofluorimetry results. Therefore, we now provide the measure of the activity of Rab11 through cellular FRET in COS-7 cells. As shown in new Figure 7E, SH3BP5 expression induce a 1.4 fold increase in Rab11 activity compared to the control condition and to the two SH3BP5 mutants. This result is comparable with the experiment performed in fluorimetry, however, as suggested by the reviewer, the fold change increase in cellular FRET for SH3BP5^{wt} is slightly higher.

Figure 7D now contains panels showing AS-Rab11 localization, and an experiment showing COS-7 cells transfected with AS-Rab11 alone.

In Figure 7E the colocalization between Rab11 activity and SH3BP5 mutants is shown. The graph displays a significant decrease in the Rab11 activity with the colocalization of SH3BP5^{AAA} and SH3BP5^{AK}, compared to SH3BP5^{wt}. As suggested by the reviewer, AS-Rab11 and SH3BP5 mutants expression could influence the respective localizations of the two proteins. We now rule out this possibility by providing the results of colocalization between SH3BP5^{wt}, SH3BP5^{AAA}, or SH3BP5^{AK} and total AS-Rab11, showing unchanged Pearson's correlation coefficient among these 3 conditions (Figure S8B).

4. The analysis of Rab11 patient mutations does not add much to the manuscript, although the assumptions are all reasonable. At least the point mutations that are suspected to effect GEF activity should be tested in vitro to substantiate this part (K13N, R82C, and A68T as a control).

We have significantly expanded experiments on the clinically relevant Rab11 mutations. We found experimentally using HDX-MS that V22M and S154L result in loss of nucleotide binding, and destabilisation of the entire Rab11 protein (Fig. S7E,F). We also carried out GEF assays on K13N, K24R, A68T, and R82C. We find that K13N located at the interface with SH3BP5 results in a greater than 100-fold decrease in GEF activity, with R82C and A68T having a minor decrease in SH3BP5 mediated exchange. The K24R mutant which directly interacts with bound nucleotide destabilises nucleotide binding, with exchange no longer requiring GEF activity. This

data has been included in Fig. 5, with additional descriptions included in the results and discussion (see lines 314-347).

Additional points: - SH3BP5 runs at an apparent MW of 78 kDa on gel filtration. Is this the result of dimerization or the v-shape? Could any of the crystallographic contacts represent a dimerization interface in solution.

None of the crystallographic contacts between SH3BP5 in the unit cell are likely to be intact in solution (CSS scores in PISA of 0.0). It is most likely that the early elution off of gel filtration is due to the extended v-shape. As gel filtration is a poor measure of MW, we have removed the MW labels from the Figure S3.

The references to Fig S3 are wrong in the text (l. 155 and 157)

This has been corrected in text.

- For the Fo-Fc omit map, side chains should be shown as well – it would be nice to see that K41 is well defined.

We have included both a feature enhanced map (FEM) and a Fo-Fc omit map for switch I and switch II with the side chains shown in Fig. S4. The maps show that only part of the K41 is ordered, with most of the side chain showing no clear density. However, even with the lack of density, K41 is still pointing towards the Mg²⁺ binding site. To indicate this ambiguity of the exact location of K41 we have modified Fig. 8, and added more nuanced discussion of this in the figure legend/discussion.

- Axis units in Fig 1C and 5B should be 10E-3

This has been modified in text.

- All shown side chains should be labeled in Fig 4A, and the font size should be increased.

We have increased the font size for the side chain labels in Fig. 4, however, including all residue labels makes this figure very difficult to interpret. As the most important information is the exact orientation of the F36/I44 residues in Rab11 relative to other Rab:GEF complexes, only these residues have been labeled. We agree that it is important to label the side chains for the Rab11 complexes to provide the full information on this interface, and we have added an additional Fig. S6 where all residues are labelled for the GDP bound Rab11 and the SH3BP5 bound Rab11.

- The comparison of Sec2 and SH3BP5 does not suggest convergent evolution, but simply independent evolution: the two GEFs mediate nucleotide exchange by different interactions/interfaces, they just happen to both be coiled-coils.

We agree with the reviewer, and have changed the wording to independent evolution of coiled coils to acts as Rab GEFs.

- A wt reference curve would be helpful in the panels of Fig 5A.

We have added in a WT reference curve to this figure (now Fig. 6 in the revised version).

- Not every data point in Fig 5B has an error bar, thus only represents a single experiment?

All experiments were carried out in triplicate, some errors bars are smaller than the size of the point. This has been clarified in the legend.

Reviewer #3 (Remarks to the Author):

The manuscript from the Burke group describes the previously unknown crystal structure of the small GTPase in complex with its cellular activator SH3BP5.

Small GTPases control crucial signaling events in the eukaryotic cell. In order to become active, they require guanine nucleotide exchange factors (GEFs) that replace the tightly bound GDP-nucleotide with GTP. Until now, the mechanism of activation of the trafficking regulator GTPase Rab11 was unknown. The authors therefore have characterized the complex of Rab11 with its GEF SH3BP5 structurally and biochemically. For the first time, the mechanism of Rab11-activation is understood. Thus, these data may be of use when analyzing Rab11 dependent intracellular transport processes.

The manuscript is well written and the data are of the highest quality. There is no doubt on the technical excellence of the study. I therefore only have a general point of criticism (see below), but I recommend the article to Nature Communications after proper amendment.

We appreciate the overall positive assessment of the manuscript.

Major comments

- page 11, from 243: The authors aim at analyzing the determinants for SH3BP5-specificity towards Rab11. To this purpose, they substitute Rab11 positions being apparently crucial for GEF-binding with the corresponding amino acids from other Rabs (L38P, S40F, K41P). However, the mutations they introduce by the approach may affect the structure of Rab11 severely, both locally and globally. This argument applies to the proline substitutions especially, since they may alter the proper conformations and positioning of the adjacent regions and may also introduce or abrogate crosstalk with other side chains. I therefore consider this approach as being inappropriate to understanding the importance of individual Rab11 amino acids or for confirming the Rab11:SH3BP5-complex interface on the basis of the crystal structure. Rather, the author should alter positions L38, S40, K41, and others into alanine and quantify the effect of GEF-mediated exchange. Alternatively, they may create SH3BP5 alanine mutants for the same purpose. Confirming the complex interface as observed in the crystal structure should be included.

However, I believe that the authors have correctly interpreted their structural data. This final biochemical analysis is simply required as an ultimate confirmation.

We agree that the mutations selected will cause differences both locally and globally. We believe the mechanism of why the L38P, S40F, and K41P can't be activated by SH3BP5 is due primarily to preventing the proper orientation of the critical F36 and I44 residues. However, we agree with the reviewer that it is important to directly verify the interface through alanine mutations. We have altered F36, L38, S40, K41, and I44 into alanine. F36 and I44 which directly interact with SH3BP5 almost completely prevent GEF mediated exchange (>100 fold decrease in SH3BP5 mediated exchange). S40A, L38, and K41 also lead to partially decreased GEF mediated exchange. The full experiments have been described in Fig. 6, with additional discussion in both the results (lines 314-347) and discussion sections of the text.

Minor comments

- General comment: I recommend that the authors add a (sub-)section describing the GEF-reaction. How is the nucleotide believed to being displaced from its binding pocket? From the manuscript I conclude that F36 should have a major impact. If this is the case, the authors should clearly mention the role of F36 in binding to the guanine base of GDP/GTP and its relevance for high nucleotide affinity. I could not find such a statement or a similar one in the main text of the manuscript.

We have added an additional sub-section describing the GEF reaction in the discussion section. We have added discussion on the role of F36 in binding nucleotide, and the key role in SH3BP5 in disrupting that interaction. This is included in lines 461-475.

- page 10, from line 220: The authors speculate on the potential consequences of disease-related Rab11 amino acid substitutions for the interaction with SH3BP5. However, not data are presented and thus this part remains rather speculative. It is advised to move the section into the discussion.

We have added significant additional experiments on the role of the clinical mutations and how they affect SH3BP5 mediated GEF activity. These experiments are now shown in Fig. 5, and are clearly described in the text. We think this makes an important contribution to the manuscript, as it shows that Rab11 clinical mutations can directly alter the association with SH3BP5.

- I recommend including a figure/supplementary figure that demonstrates the most important amino acid interactions of the Rab11-SH3BP5 interface (in particular F36 and I41 of Rab11). This will facilitate understanding the crucial involvement of the respective side chains for the complex formation.

This is an excellent suggestion by the reviewer. We have added a panel to Figure 6 in the mutant panel that clearly highlights the residues and their respective roles in complex formation, as well as an additional supplemental figure (Fig. S6) that clearly compares the orientation of these residues between GDP bound Rab11 and SH3BP5 bound Rab11.

- author statement “Intriguingly, dominant negative mutants of 135 Rab11A (S25N) caused rapid release of mant-GDP even in the absence of SH3B”: It is well known that the mutation affects Mg²⁺ coordination and therefore decreases nucleotide affinity (i.e. increases dissociation rate of

the nucleotide). Since the finding is not surprising, please amend accordingly (or delete the sentence since the finding is of no added value for the manuscript).

We agree that this is an unsurprising finding, and we have reworded this accordingly.

- the supplementary movie is a good visual aid to the reader. Nevertheless, the authors should try to improve a) the definition of the helices. In particular, the N- and C-terminal ends of the helices appear to be defined awkwardly. Some regions are depicted as helices but are obviously not.

Unfortunately, representing a structure as cartoon helices in pymol during a morph between two states is a technically daunting task. This is because it picks the secondary structure that exists in either the starting or ending state and maintains them through the movie. To fix this problem we have now indicated Rab11 as a ribbon representation in the movie, as this removes complications of secondary structure differences between states.

b) the bond geometry of the side chains. For instance, the phenyl ring of the switch 1 phenylalanine is apparently being distorted during the morph.

The movie does not represent a molecular dynamics trajectory between states, it represents a morph in pymol, which moves the atoms between the two states in a linear trajectory. The geometry of the intermediate states between the beginning and end are highly distorted, and this is a consequence of how the movie is generated. The movie is not supposed to indicate the exact conformational changes that occur, but is merely a visual aid to assist the reader in interpreting the differences between nucleotide bound and SH3BP5 bound. How the movie was generated and its limitations has been more clearly described in the figure legend.

REVIEWERS' COMMENTS:

Reviewer #1 (Remarks to the Author):

The authors addressed all my concerns; therefore I recommend publication of this manuscript.

Reviewer #2 (Remarks to the Author):

The authors did an excellent job addressing my questions and suggestions. They present a nice story that I recommend for publication without any reservations.